



# The microbial community model MCoM 1.0: A scalable framework for modelling phototroph-heterotrophic interactions in diverse microbial communities

Leonhard Lücken[1], Michael J. Follows[2], Jason G. Bragg[3], and Sinikka T. Lennartz[1]

[1]Institute for Chemistry and Biology of the Marine Environment (ICBM), Carl von Ossietzky Universität Oldenburg, Germany
[2]Department of Earth, Atmospheric and Planetary Sciences, Massachusetts Institute of Technology, Cambridge, MA, United States
[3]Research Centre for Ecosystem Resilience, Botanic Gardens of Sydney, Sydney, New South Wales, 2000, Australia

**Correspondence:** Leonhard Lücken (leonhard.luecken@uol.de)

**Abstract.** Microbial communities, comprising phototrophic and heterotrophic microorganisms, play a crucial role in global biogeochemical cycles. However, existing biogeochemical models rarely take into account the complex interactions between these groups, usually focusing on competition for resources. In this work, we introduce the Microbial Community Model (MCoM), a framework for simulating the dynamics of diverse microbial communities. MCoM incorporates a wide range of interactions, such as cross-feeding, metabolite effects, and competition for nutrients. The model differentiates between dissolved organic nutrients (DON) and carbon (DOC), accounts for phytoplankton and heterotrophic bacterial species-specific organic matter release and uptake profiles, and simulates the impacts of bacterial metabolites on phytoplankton growth. Implemented as a box model, MCoM tracks phototrophic and heterotrophic biomass, active metabolites, DOC, DON, and inorganic nutrients through non-linear differential equations, enabling the exploration of emergent properties and feedbacks. We demonstrate the model's capabilities through simulations of experimental data of pairwise co-cultures of heterotrophic and phototrophic microorganisms, and find overall good agreement. Due to the scalable implementation, interaction matrices for larger, i.e. hundreds, of microbial groups can easily be realised. We show examples for such applications of MCoM in assessing emergent dynamics, including periodic succession patterns and multi-stability. MCoM provides a versatile, scalable, and customizable platform for assessing the range from pairwise interactions to complex microbial communities and their impact on biogeochemical fluxes.

## 1 Introduction

Microorganisms are the main driver of global biogeochemical cycles of major elements such as carbon, nitrogen and phosphorus. The microbially mediated turnover of these elements does not happen in isolation, because microorganisms live in diverse, interacting communities. The study of such communities in oceanic ecosystems and their adequate representation in biogeochemical models is key to understanding global elemental cycles, which in turn are crucial for the Earth's ecosystems and the habitability of the planet (Tagliabue, 2023). Arguably the two most fundamental roles in these ecosystems are primary





producers that fix inorganic carbon and nutrients into organic biomass by photosynthesis, and their counterparts, heterotrophic organisms that decompose organic matter back to $CO_2$ and inorganic nutrients. Biogeochemical models are powerful tools to quantify elemental fluxes mediated by these microorganisms at various scales. They can generally achieve good agreement with observations in terms of common macronutrients and carbon reservoirs (e.g., Friedrichs et al., 2007; Séférian et al., 2012), and a biogeochemical component is a central part of state-of-the-art global Earth System Models.

Interactions between phototrophic and heterotrophic microorganisms are widespread, and can comprise positive and negative interactions at the same time (Cirri and Pohnert, 2019; Kost et al., 2023). On one hand, the growth rates of heterotrophic consumers are primarily driven by the availability of organic matter, which is synthesized from inorganic nutrients and carbon by phytoplankton populations. A key factor influencing the composition of the heterotroph community is the specific composition of the organic material produced by phytoplankton, which varies between species (Sarmento et al., 2013; Becker et al., 2014; Mühlenbruch et al., 2018) and favors different consumers that are adapted to specific compounds (Teeling et al., 2012; Sarmento et al., 2013; Elovaara et al., 2021). Phytoplankton not only release DOM through the breakdown of previously assimilated biomass but also exude excess of newly fixed carbon (Myklestad, 2000; Thornton, 2014). The composition of these exudates can vary significantly depending on environmental conditions, phytoplankton species, and growth phase, further influencing the dynamics of heterotrophic bacterial communities (Buchan et al., 2014), and the elemental ratios of this organic matter is influenced by the factor limiting their growth, resulting in varying carbon-to-nutrient ratios (Saad et al., 2016). On the other hand, phototrophs rely on the recycling of organic matter that provides new nutrients, a process mediated by heterotrophic organisms.

A main limitation of traditional biogeochemical modelling approaches is their focus on competition for resources among microbes, mainly phytoplankton, neglecting widespread experimental evidence that interactions between phototrophs and heterotrophs comprise positive, neutral and negative interactions resulting from cross-feeding or metabolite exchange (Sher et al., 2011; Seymour et al., 2017; Cirri and Pohnert, 2019). For example, certain bacteria release siderophores, which can either sequester iron, limiting its availability to phytoplankton or make it more accessible (Maldonado and Price, 1999; Kazamia et al., 2018). Other bacteria produce algicides (Coyne et al., 2022), while some synthesize beneficial metabolites such as vitamin B12 (Sultana et al., 2023), or hormones like auxins (Amin et al., 2015). Further, heterotrophic bacteria may relieve oxidative stress on phytoplankton by degrading reactive oxygen species, e.g., hydrogen peroxide (Morris et al., 2022).

These microbial interactions are ubiquitous, and systematically affect the rates of growth and elemental turnover of microorganisms and, hence, biogeochemical fluxes (Seymour et al., 2017). Even though these interactions occur on cellular scales, they may have cascading effects on the entire ecosystem with consequences for carbon cycling and, ultimately, climate regulation. However, a mathematical framework on how to incorporate these widespread microbial interactions and link it to elemental turnover is missing, hampering our ability to systematically assess their community-level effects. Existing theoretical approaches have started including cooperation (De Mazancourt and Schwartz, 2010) or facilitation (Koffel et al., 2021) into consumer-resource theory, but have not yet been applied in biogeochemical modelling. To address the middle ground between purely theoretical and species specific models, we present the Microbial Community Model (MCoM), which is designed to simulate the dynamics of microbial communities, encompassing both phototrophic and heterotrophic populations and account



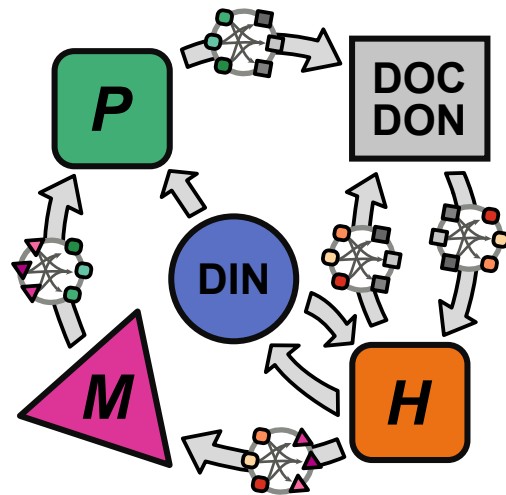

**Figure 1.** Model structure as flux diagram between different pools included in MCoM: phototrophic ($P$) and heterotrophic ($H$) populations, DOM (dissolved organic carbon, DOC, and nutrient, DON), metabolites ($M$), and inorganic nutrient (DIN). Schematic networks on links indicate complex connectivities between the components of the aggregate pools. Phytoplankton populations consume dissolved inorganic nutrients, are affected by bacterial metabolites, and release dissolved organic matter. Heterotrophic bacteria consume and release (differently composed) DOM, and consume or produce inorganic nutrient, depending on the availability of DON. Further, heterotrophs may release metabolites affecting phytoplankton growth.

for various types of interactions. MCoM incorporates features such as the differentiation of dissolved organic nutrients (DON) and carbon (DOC), DOM release profiles of phytoplankton and heterotrophic bacterial species, DOM preference profiles of heterotrophs, as well as the positive or negative impacts of bacterial products on phytoplankton growth. Additionally, the

60 model accounts for competition for inorganic nutrients, excess production of DON and DOC by phytoplankton, and excess remineralization of DON by heterotrophic bacteria. Currently implemented as a box model, MCoM tracks state variables such as biomass, DOC, DON, and inorganic nutrients in a homogeneous volume of water through non-linear differential equations. This approach allows exploration of emergent properties and feedbacks within microbial communities, which may be essential for advancing predictive oceanic and climate modelling.

We describe the mathematical formulation and numerical implementation of the model in the Sections 2 and 3. In Section 4, we present several examples to evaluate its capabilities to fit experimental data and to describe emergent dynamics.

## 2 Model description

In MCoM, any momentary state of the microbial model system is represented by the following dynamic variables:

– Phytoplankton population densities $P_i(t)$, $i \in \mathcal{P}$ (cells L$^{-1}$)



– Heterotroph population densities $H_i(t)$, $i \in \mathcal{H}$ (cells L$^{-1}$)

    – Concentrations of dissolved organic nutrient (DON) compounds $D_i(t)$, $i \in \mathcal{N}$ (mM(N) [= mmol(N) m$^{-3}$])

    – Concentrations of dissolved organic carbon (DOC) compounds $D_i(t)$, $i \in \mathcal{C}$ (mM(C) [= mmol(C) m$^{-3}$])

    – Concentrations of bacterial metabolites acting on phytoplankton growth $M_i(t)$, $i \in \mathcal{M}$ (L$^{-1}$)

    – The nutrient concentration $N(t)$ (mM(N))

We are using index sets $\mathcal{P}$, $\mathcal{H}$, $\mathcal{N}$, $\mathcal{C}$, and $\mathcal{M}$ throughout the text to enumerate the different components of a specific type. For the microbial populations, we assume fixed stoichiometric molar ratios $r_i^{C:N}$ (for $i \in \mathcal{P}$ and $i \in \mathcal{H}$) and an average cell carbon content $\chi_i^C$. For an overview of parameters and associated physical units, we refer to Table A1. The populations are measured in cells per litre, but elemental C- and N-content can be associated unambiguously given $r_i^{C:N}$ and $\chi_i^C$. The dynamics of a component is determined by the sum of its associated in- and outfluxes, which are denoted by $f_X^Y$ with varying super- and

subscripts (*cf.* Table A2). Figure 1 summarizes these fluxes and provides a schematic overview of the model. In the following, we describe the different involved processes in detail.

## 2.1 Phytoplankton population dynamics

The growth of a phytoplankton population $P_i$ is governed by

$$\frac{d}{dt}P_i = \underbrace{\left(f_{C\to i}\right.}_{\text{biomass assimilation}} \underbrace{\left.- f_{i\to\text{DOC}}\right)}_{\text{biomass decay}}/\chi_i^C - \underbrace{\Delta P_i,}_{\text{dispersion}} \tag{1}$$

where $\chi_i^C$ is the cellular carbon content, $f_{C\to i}$ is the photosynthetic carbon assimilation flux, and $f_{i\to\text{DOC}}$ summarizes losses of biomass to DOC. Assuming a fixed stoichiometric composition $r_i^{C:N}$, nutrient fluxes must fulfil

$$\underbrace{f_{C\to i} = r_i^{C:N} \cdot f_{N\to i},}_{\text{influx}} \text{ and } \underbrace{f_{i\to\text{DOC}} = r_i^{C:N} \cdot f_{i\to\text{DON}}.}_{\text{outflux}} \tag{2}$$

We apply Liebig's minimum principle to distinguish two different growth regimes, depending on the limiting factor. Specifically, light intensity determines the maximal carbon assimilation rate $f_{C\to i}^{\max}$, while nutrient availability governs the maximal

nutrient assimilation rate $f_{N\to i}^{\max}$. As the realized fluxes must adhere to the stoichiometric constraints (2), nutrient limitation occurs when

$$r_i^{C:N} \cdot f_{N\to i}^{\max} < f_{C\to i}^{\max}, \tag{3}$$

and light limitation when $r_i^{C:N} \cdot f_{N\to i}^{\max} > f_{C\to i}^{\max}$.




### 2.1.1 Maximal and realized assimilation rates

Under conditions where nutrient availability is the primary limiting factor, nutrient uptake is defined by the nutrient concentration ($f_{N\to i} = f_{N\to i}^{\max}$) and determines phytoplankton growth. This maximal uptake is assumed to be characterized by a type-II response function:

$$f_{N\to i}^{\max} = \frac{V_{i,N} N}{K_i^N + N} P_i, \tag{4}$$

where $\chi_i^N$ is the nutrient content per cell, $V_{i,N}$ is the (theoretical) maximum nutrient uptake rate (different to $f_{N\to i}^{\max}$, which is the maximal rate under current conditions) and $K_i^N$ is the corresponding half-saturation constant. The corresponding carbon assimilation $f_{C\to i}$ is calculated according to Eq. (2).

In light-limited regimes the growth is governed by the maximal photosynthetic carbon assimilation

$$f_{C\to i}^{\max} = \phi_i(I)\,\chi_i^{\mathrm{Chl}} P_i,$$

where $\chi_i^{\mathrm{Chl}}$ is the average chlorophyll content per cell and $\phi_i(I)$ is the P-I curve associated to phytoplankton species $P_i$, which describes the dependence of the photosynthesis rate on the irradiance $I$. For the P-I curve we assume the form

$$\phi_i(I) = \phi_{s,i}\left(1 - \exp\left(\frac{-\alpha_i \cdot I}{\phi_{s,i}}\right)\right) \cdot \exp\left(\frac{-\beta_i \cdot I}{\phi_{s,i}}\right) \tag{5}$$

with initial sensitivity $\alpha_i$, photoinhibition coefficient $\beta_i$ and upper bound $\phi_{s,i}$ for the photosynthesis rate (Platt et al., 1980).

### 2.1.2 Assimilation and exudation of DOM

Phytoplankton species exude excessively fixed organic matter directly into the environment (Myklestad, 2000; Thornton, 2014). In the nutrient limited regime the surplus photosynthetic capacity is assumed to be used for exudation of DOC, which is calculated as

$$f_{C\to \mathrm{DOC}}^i = f_{C\to i}^{\max} - f_{C\to i}. \tag{6}$$

In the light limited regime, exudates are assumed to be nutrient-saturated[1] and we employ an extended scheme to determine the exudation rates, because organic exudates contain a minimal amount of carbon. We assume that nutrient-rich DOM has a uniform C:N ratio $r_{\mathrm{ex},i}^{C:N}$ (with $r_{\mathrm{ex},i}^{C:N} < r_i^{C:N}$), i.e.,

$$f_{C\to \mathrm{DOC}}^i = r_{\mathrm{ex},i}^{C:N} \cdot f_{N\to \mathrm{DON}}^i. \tag{7}$$

Notably, this assumption implies that not all carbon is utilized for biomass assimilation, even though its fixation rate limits the population's growth. To maintain a balance between growth and exudation in nutrient-rich environments, we introduce a maximal fraction $q_i^{\mathrm{ex}}$ of $f_{C\to i}^{\max}$, up to which carbon can be allocated for exudation. A maximization of DON exudation under

[1]Light-limited exudation can also be switched off by setting variant.use_exudation=false.





this constraint (see Appendix B) gives

$$f_{C\rightarrow \text{DOC}}^i = \min\left(\frac{r_{\text{ex},i}^{C:N}}{r_i^{C:N} - r_{\text{ex},i}^{C:N}}\left(r_i^{C:N} f_{N\rightarrow i}^{\max} - f_{C\rightarrow i}^{\max}\right), q_i^{\max} f_{C\rightarrow i}^{\max}\right). \tag{8}$$

Figure 2 shows the resulting dependence of assimilation and exudation on nutrient availability with fixed photosynthesis rate $f_{C\rightarrow i}^{\max}$. For low nutrient concentrations, the growth rate follows the maximal nutrient uptake rate, i.e., $f_{C\rightarrow i} = r_i^{C:N} f_{N\rightarrow i}^{\max}$ (overlapping black and blue curves), *cf.* Eq. (4) and only DOC is excreted at a rate $f_{C\rightarrow DOC}^i$ (dashed gray curve). The growth rate reaches its maximum around a concentration $N = 0.4\,\text{mM}$, where $r_i^{C:N} f_{N\rightarrow i}^{\max} = f_{C\rightarrow i}^{\max}$ and no DOM is excreted. For larger concentrations, the growth becomes light-limited and excreted DOM has C:N ratio $r_{\text{ex},i}^{C:N}$. Within the interval $0.4 < N < 0.47$ (shaded background), the first argument of the minimum function in Eq. (8) takes effect, and the fraction of photosynthesised organic carbon allocated to exudation is smaller than $q_i^{\max}$. For increasing nutrient an increasing share of fixed organic carbon is exuded until the share reaches the maximal value of $q_i^{\max}$ at $N \approx 0.47$ and the exudation profile becomes independent of further increases in nutrient concentrations.

### 2.1.3 Population losses

The phytoplankton mortality is governed by the sum of three terms:

$$f_{i\rightarrow \text{DOC}} = \chi_i^C \cdot \max\left(0, \Big[\underbrace{\delta_i}_{\text{linear mortality}} + \underbrace{\delta_{q,i} P_i}_{\text{quadratic mortality}} + \underbrace{\sum_{j\in\mathcal{M}} \frac{a_{i,j} M_j}{h_{i,j} + M_j}}_{\text{metabolite production}}\Big] P_i\right), \tag{9}$$

where $\delta_i$ is the base rate of linear mortality, $\delta_{q,i}$ is the coefficient of quadratic mortality, and the sum $\sum_{j\in\mathcal{M}} \frac{a_{i,j} M_j}{h_{i,j} + M_j}$ describes the total impact of present metabolites on $P_i$. Note that the second term in the maximum function may theoretically become negative, because the coefficients $a_{i,j}$ are allowed to be negative in order to model beneficial metabolite effects. To ensure that such a mortality reduction does not result in a positive growth, the maximum ensures the positivity of the flow $f_{i\rightarrow \text{DOC}}$.

## 2.2 Heterotroph population dynamics

The bacterial population densities $H_i$ change according to:

$$\frac{d}{dt}H_i = \underbrace{\Big(f_{\text{DOC}\rightarrow i}}_{\text{biomass assimilation}} - \underbrace{f_{i\rightarrow \text{DOC}}\Big)}_{\text{biomass decay}}/\chi_i^C - \underbrace{\Delta H_i}_{\text{dispersion}}, \tag{10}$$

where $f_{\text{DOC}\rightarrow i}$ describes the carbon assimilation into biomass from DOC compounds and $f_{i\rightarrow \text{DOC}}$ the loss of biomass to organic compounds. As for phytoplankton, heterotrophs are modelled with a fixed stoichiometric composition $r_i^{C:N}$, which is preserved by imposing

$$\underbrace{f_{\text{DOC}\rightarrow i} = r_i^{C:N}\left(f_{\text{DON}\rightarrow i} + f_{N\rightarrow i}\right)}_{\text{influx}}, \text{ and } \underbrace{f_{i\rightarrow \text{DOC}} = r_i^{C:N} \cdot f_{i\rightarrow \text{DON}}}_{\text{outflux}}. \tag{11}$$





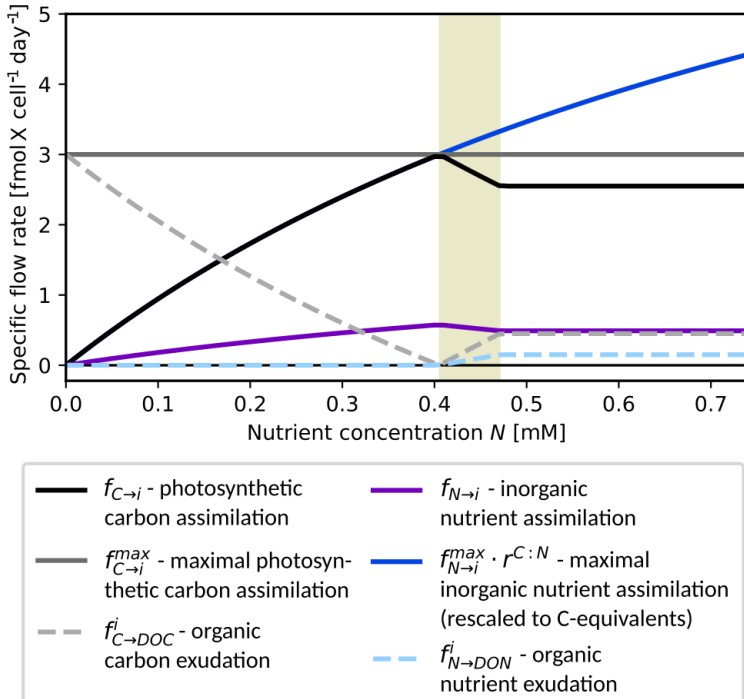

**Figure 2.** Excess DOM exudation by phytoplankton. Exudation rates $f_{C\rightarrow DOC}$ (gray dashed) and $f_{N\rightarrow DON}$ (light blue dashed) are plotted against nutrient availability $N$ for a fixed photosynthesis rate $f_{C\rightarrow i}^{\max} = 3.0\,\text{fmol(C)}\,\text{cell}^{-1}\,\text{day}^{-1}$ (dark gray) and a single cell, i.e., $P_i = 1$. Additional curves: (rescaled) maximal nutrient assimilation rate $f_{N\rightarrow i}^{\max}(N) \cdot r_i^{C:N}$ (blue), realized assimilation rates $f_{C\rightarrow i}$ (black) and $f_{N\rightarrow i}$ (purple). Other parameters for this example: $r_i^{C:N} = 5.2\,\text{mol(C)}\,\text{mol(N)}^{-1}$, $r_{\text{ex}}^{C:N} = 3.0\,\text{mol(C)}\,\text{mol(N)}^{-1}$, $q_i^{\max} = 0.15$, $V_{i,N} = 2.0\,\text{day}^{-1}$, $K_i^N = 1.0\,\text{mM(N)}$.

Since heterotrophic populations are assumed to obtain nutrients from inorganic and organic sources, the constraint for the influx involves both sources. Importantly, the DOM assimilation fluxes are aggregates of fluxes from individual compounds, i.e.,

$$f_{\text{DOC}\rightarrow i} = \sum_{j\in\mathcal{C}} f_{j\rightarrow i}, \text{ and } f_{\text{DON}\rightarrow i} = \sum_{j\in\mathcal{N}} f_{j\rightarrow i}. \tag{12}$$

Growth may be limited either by DOC or by total (organic and inorganic) nutrient availability, and the realized assimilation

flux is determined by a minimum principle comparing maximal total assimilation rates of carbon ($f_{\text{DOC}\rightarrow i}^{\max}$) and nutrient ($f_{N\rightarrow i}^{\max} + f_{\text{DON}\rightarrow i}^{\max}$). A population $H_i$ is nutrient limited if

$$r_i^{C:N}\left(f_{N\rightarrow i}^{\max} + f_{\text{DON}\rightarrow i}^{\max}\right) < f_{\text{DOC}\rightarrow i}^{\max}, \tag{13}$$

and energy limited if the opposite holds.





### 2.2.1 Maximal assimilation rates

The maximal uptake rate of an individual organic compound $D_j$ by population $H_i$ is modelled as

$$f_{j\to i}^{\text{up,max}} = \frac{V_{i,j} D_j}{K_{i,j} + D_j} H_i, \text{ for } j \in \mathcal{N} \text{ or } j \in \mathcal{C}. \tag{14}$$

Here, $V_{i,j}$ is the (theoretical) maximal uptake rate [*cf.* Eq (4)] and $K_{i,j}$ is the corresponding half-saturation constant. Similarly, the maximal uptake of inorganic nutrient is

$$f_{N\to i}^{\max} = \frac{V_{i,N} N}{K_i + N} H_i. \tag{15}$$

When consuming DOM compounds, a part of the compounds is assumed to be catabolized, i.e., metabolically degraded for energy extraction. Hence, only a part of the uptake is available for integration into biomass and the remainder is remineralized and released into the inorganic nutrient and carbon pools. Correspondingly, the maximal assimilation rate for the compound $D_j$ is calculated as

$$f_{j\to i}^{\max} = Y_{i,j} f_{j\to i}^{\text{up,max}}, \tag{16}$$

where the yield coefficient $Y_{i,j}$ determines the assimilated fraction. The resulting maximal organic assimilation fluxes are

$$f_{\text{DOC}\to i}^{\max} = \sum_{j \in \mathcal{C}} Y_{i,j} f_{j\to i}^{\text{up,max}}, \text{ and, } f_{\text{DON}\to i}^{\max} = \sum_{j \in \mathcal{N}} Y_{i,j} f_{j\to i}^{\text{up,max}}. \tag{17}$$

### 2.2.2 DOM assimilation and remineralization

If the population $H_i$ grows under nutrient limited conditions, it takes up and assimilates DON compounds at the maximum possible rates, i.e.,

$f_{j\to i} = Y_{i,j} f_{j\to i}^{\text{up,max}}, \text{ for } j \in \mathcal{N}. \tag{18}$

The fraction of DON uptake, which is not assimilated, is remineralized and induces a flow from the organic pool $D_j$ to the inorganic nutrient pool $N$:

$$f_{j\to N}^{i} = (1 - Y_{i,j}) f_{j\to i}^{\text{up,max}}. \tag{19}$$

Further, the inorganic nutrient is assimilated at maximum rate $f_{N\to i} = f_{N\to i}^{\max}$ as well. Adhering to the stoichiometric con-

175 straints (11), the uptake of DOC is regulated down. We assume that this regulation decreases the uptake of all available DOC compounds proportionally, i.e., for $j \in \mathcal{C}$, $f_{j\to i}^{\text{up}} = c \cdot f_{j\to i}^{\text{up,max}}$ with a factor $c = r_i^{C:N} (f_{N\to i}^{\max} + f_{\text{DON}\to i}^{\max}) / f_{\text{DOC}\to i}^{\max}$, yielding corresponding assimilation and remineralization fluxes of

$$f_{j\to i} = Y_{i,j} f_{j\to i}^{\text{up}}, \text{ and } f_{j\to C}^{i} = (1 - Y_{i,j}) f_{j\to i}^{\text{up}}. \tag{20}$$

In carbon limited situations [Eq. (3) is not fulfilled], DOC is used at maximum rates, i.e., for all $j \in \mathcal{C}$:

$f_{j\to i}^{\text{up}} = f_{j\to i}^{\text{up,max}}, \text{ and } f_{j\to i} = Y_{i,j} f_{j\to i}^{\text{up,max}}. \tag{21}$





While the total assimilation rate of nutrient is tied to $f_{\text{DOC}\to i}$ by stoichiometry, MCoM implements two variants of how heterotrophs behave with regard to excess DON uptake capacities.[2] Either, (i) heterotrophs still take up maximal amounts of DON and eventually remineralize any excess, or (ii) they homogeneously decrease the uptake rates for the different DON compounds such that nutrient assimilation matches carbon assimilation. The latter regulation is implemented analogously as above for DOC compounds in the nutrient limited case. In any case, we assume that DON is used preferentially and inorganic nutrient is only taken up to a degree necessary to serve the nutrient demands. More explicitly, for variant (i) we assume that $f_{j\to i}^{\text{up}} = f_{j\to i}^{\text{up,max}}$ for $j \in \mathcal{N}$. If the DON uptake exceeds DOC uptake, i.e.,

$$r_i^{C:N} f_{\text{DON}\to i}^{\max} > f_{\text{DOC}\to i}^{\max}, \tag{22}$$

additional remineralization occurs:

$$f_{\text{DON}\to N}^{+} = f_{\text{DON}\to i}^{\max} - f_{\text{DOC}\to i}^{\max}/r_i^{C:N}. \tag{23}$$

If Eq. (22) holds for variant (ii), the realized uptake rates for DON compounds are modified as $f_{j\to i}^{\text{up}} = c \cdot f_{j\to i}^{\text{up,max}}$ with a factor $c = f_{\text{DOC}\to i}^{\max}/\left(r_i^{C:N} f_{\text{DON}\to i}^{\max}\right)$ and $f_{\text{DON}\to N}^{+} = 0$.

Figure 3 shows the uptake, remineralization, as well as assimilation for varying DON at fixed DOC and inorganic nutrient concentrations. For growth limiting concentrations of DON, DON availability controls the population growth rate and is remineralized at a minimal fraction. For intermediate concentrations, the fraction of remineralized DON remains the same but less and less inorganic nutrient is used (shaded interval $0.33 < \text{DON} < 0.47$). For DON concentrations above $0.47\,\text{mM}$, all nutrient requirements are served by DON and for increasing concentrations, the surplus DON is completely remineralized [variant (i), solid light blue curve] or DON remineralization rates remain constant for higher DON concentrations [variant (ii), dashed light blue curve].

### 2.2.3 Population loss and metabolite investment

Heterotroph biomass loss is assumed to consist of three components:

$$f_{i\to\text{DOC}} = \chi_i^C \cdot \left(\underbrace{\delta_i H_i}_{\text{linear mortality}} + \underbrace{\delta_{q,i} H_i^2}_{\text{quadratic mortality}}\right) + \underbrace{\Pi_i.}_{\text{metabolite production}} \tag{24}$$

Here, $\delta_i$ is the base rate of linear mortality, $\delta_{q,i}$ is the quadratic mortality coefficient, and

$$\Pi_i = \sum_{j\in\mathcal{M}} \pi_{j,i} f_{\text{DOC}\to i} \tag{25}$$

is the metabolite production investment with fractions $\pi_{j,i}$ of newly assimilated biomass expended for the production of metabolite $M_j$.

---

[2]This behaviour is controlled by the parameter variant.surplus_remineralization.





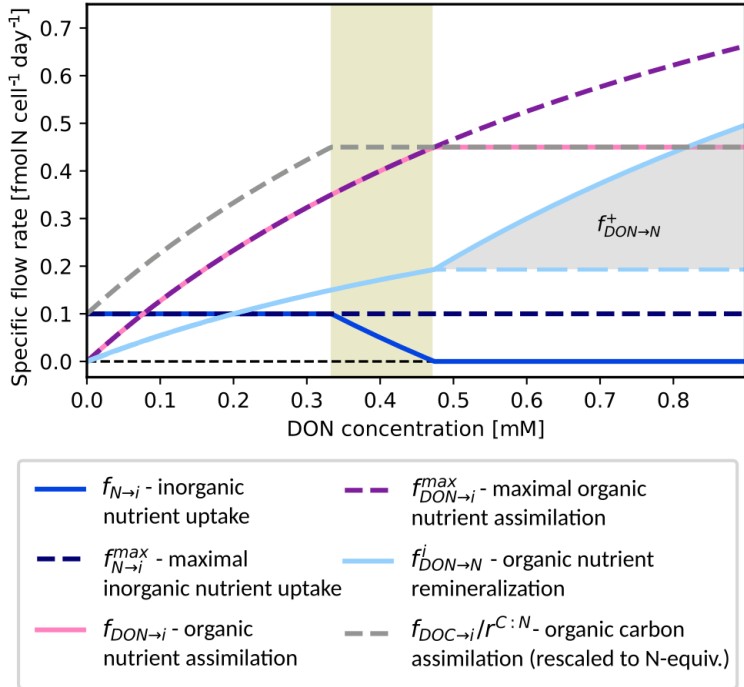

**Figure 3.** Bacterial remineralization of DON by one cell of a population $H_i$. Remineralization rate $f^i_{\text{DON}\to N}$ (light blue) plotted against DON availability at constant nutrient and DOC concentrations. Variants with (solid) and without (dashed) differ for DON levels ($> 0.47\,\text{mM}$). Additional curves: assimilation of DOC (rescaled), DON and DIN $f_{\text{DOC}\to i}/r^{C:N}_i$, $f_{\text{DON}\to i}$, and $f_{N\to i}$, maximal nutrient and DON assimilation $f^{\max}_{N\to i}$ and $f^{\max}_{\text{DON}\to i}$, and remineralization $f^i_{\text{DON}\to N}$. Parameters for this example: $r^{C:N}_i = 0.4$, $V_{i,\text{DON}} = 2.0$, $K_{i,\text{DON}} = 1.0$, $Y_{i,\text{DON}} = 0.7$. Maximal nutrient and carbon assimilation rates: $f^{\max}_{N\to i} = 0.1$ and $f^{\max}_{\text{DOC}\to i} = 1.8$.

## 2.3 DOM dynamics

The dynamics of individual DOM compounds are considered to follow the form

$$\frac{d}{dt}D_j = f_{H\to j} + f_{P\to j} + f_{C\to j} - f_{j\to H} - f_{j\to C} - \Delta D_j, \ \text{ for } j \in \mathcal{C}, \tag{26a}$$

$$210 \quad \frac{d}{dt}D_j = \underbrace{f_{H\to j} + f_{P\to j}}_{\text{population decay}} + \underbrace{f_{N\to j}}_{\text{exudation}} - \underbrace{f_{j\to H} - f_{j\to N}}_{\text{uptake}} - \underbrace{\Delta D_j}_{\text{dispersion}}, \ \text{ for } j \in \mathcal{N}. \tag{26b}$$

Here, influxes originate from phytoplankton and heterotroph population decay and from DOM exudation by phytoplankton, and outfluxes correspond to consumption by heterotrophs.



In Secs. 2.1 and 2.2, we introduced the aggregate fluxes $f_{i\to\text{DOC}}$ and $f_{i\to\text{DON}}$, $i \in \mathcal{P}$ or $i \in \mathcal{H}$. To determine their impact on
the concentrations on individual DOM compounds $D_j$, we assume specific partitioning coefficients

$$\sum_{j\in\mathcal{C}} \text{R}_{j,i} = \sum_{j\in\mathcal{N}} \text{R}_{j,i} = 1. \tag{27}$$

That is, the individual fluxes are calculated as

$$f_{i\to j} = \text{R}_{j,i} f_{i\to\text{DOC}}, \text{ for } j \in \mathcal{C}, \tag{28a}$$

$$f_{i\to j} = \text{R}_{j,i} f_{i\to\text{DON}}, \text{ for } j \in \mathcal{N}. \tag{28b}$$

For simplicity, the excess DOC- or DON-exudation $f^i_{C\to\text{DOC}}$ and $f^i_{N\to\text{DON}}$ by phytoplankton is assumed to follow the same
partitioning to individual compounds:

$$f^i_{C\to j} = \text{R}_{j,i} f^i_{C\to\text{DOC}}, \text{ for } j \in \mathcal{C}, \tag{29a}$$

$$f^i_{N\to j} = \text{R}_{j,i} f^i_{N\to\text{DON}}, \text{ for } j \in \mathcal{N}. \tag{29b}$$

Further, we denote the total uptake and remineralization fluxes per compound by

$$f_{j\to N} = \sum_{i\in\mathcal{H}} f^i_{j\to N}, \; f_{j\to C} = \sum_{i\in\mathcal{H}} f^i_{j\to C}, \; f_{j\to H} = \sum_{i\in\mathcal{H}} f_{j\to i}. \tag{30}$$

### 2.4   Metabolite dynamics

Metabolite pools $M_j$ are modelled without any specific stoichiometry, since their contribution to the total DOM is assumed to
be negligible. The energy expenditure for their synthesis is modelled by the corresponding loss term in Eq. (24). The pseudo-
concentrations $M_j$ represent a basis to calculate the impact of metabolite $M_j$ on susceptible phytoplankton species as described
in Sec. 2.1. They are governed by production and decay:

$$\frac{d}{dt} M_j = \underbrace{\theta_j \sum_{i\in\mathcal{H}} \pi_{j,i} f_{\text{DOC}\to i}}_{\text{production}} - \underbrace{\delta_j M_j}_{\text{decay}} - \underbrace{\Delta M_j}_{\text{dispersion}} . \tag{31}$$

Here, the coefficient $\theta_j$ determines the amount of biomass required to synthesize a unit of $M_j$ and $\delta_j$ determines the decay rate
of $M_j$.

### 2.5   Nutrient dynamics

The change of nutrient concentration is driven by uptake from phytoplankton and heterotrophs and remineralization of DON
by heterotrophs. Further, MCoM permits to define an external nutrient concentration $N_\text{ext}$ to model a diffusive exchange with





an external domain:

$$240 \quad \frac{d}{dt}N = \underbrace{f_{\mathrm{DON}\to N}}_{\text{remineralization}} - \underbrace{(f_{N\to P} + f_{N\to H})}_{\text{consumption}} - \underbrace{\Delta\left(N - N_{\mathrm{ext}}\right)}_{\text{dispersion}}. \tag{32}$$

Here, the aggregate flows are defined as

$$f_{\mathrm{DON}\to N} = \sum_{i\in\mathcal{H}}\sum_{j\in\mathcal{N}} f^i_{j\to N}, \ f_{N\to P} = \sum_{i\in\mathcal{P}} f_{N\to i}, \text{ and } f_{N\to H} = \sum_{i\in\mathcal{H}} f_{N\to i}. \tag{33}$$

### 2.6 Scaling up to diverse microbial communities

A key feature of MCoM is the straightforward scalability to larger interaction networks. The size and connectivity of the
245 interaction network, i.e., the number of phototroph populations, heterotroph populations, metabolites, and organic compounds
can be specified in the configuration file, together with respective parameter values and interaction matrices. MCoM offers
different built-in ways to define microbial communities and their interaction networks: For controlled set-ups, it is possible
to define all rates and interactions explicitly. This explicit definition allows the user to write their own algorithms for the
generation of the community's interaction network. To generate randomized networks, the entire community or a subset of
250 parameters can be determined stochastically. When employing stochastic network generation, reproducibility can be ensured
by setting a seed for the random generation.[3] For instance, the user can quickly set up a community by providing the number
of phototrophs, heterotrophs, resources and metabolites, specifying the in- and output of populations (the number of produced
resources and metabolites, as well as the number of consumed resources in case of heterotrophs and the number of metabolites
affecting the population in case of phototrophs) and value ranges for the other system parameters. A specific option[4] allows
to define a community of phototroph-heterotroph pairs, which only interact by exchange of DOM and metabolites within the
pairs, i.e., all adjacency matrices are diagonal. Additional functions to generate interaction matrices according to customized
requirements can easily be implemented.

## 3 Implementation and performance

### 3.1 Integration scheme

Starting from a given initial state $\boldsymbol{X}_0 = (\boldsymbol{P}_0, \boldsymbol{H}_0, \boldsymbol{D}_0, \boldsymbol{M}_0, N_0)$, MCoM generates trajectories using an Adams-Bashforth
explicit two-step method (Butcher, 2005). This method generates state approximations at equidistant time points $t_n = t_0 + n \cdot dt$
using the formula

$$\boldsymbol{X}_{n+2} = \boldsymbol{X}_{n+1} + \left[\frac{3}{2}f\left(t_{n+1}, \boldsymbol{X}_{n+1}\right) - \frac{1}{2}f\left(t_n, \boldsymbol{X}_n\right)\right]dt, \tag{34}$$

where $\boldsymbol{X}_n$ is the vector containing the system state at time $t_n$. As any method, this can lead to inaccuracies when using larger
time steps. In case that a solution seems to show numerical artifacts it advisable to compare the behaviour to a solution obtained
with smaller step width.

---

[3] Parameter run.system_seed

[4] Parameter variant.competing_pairs



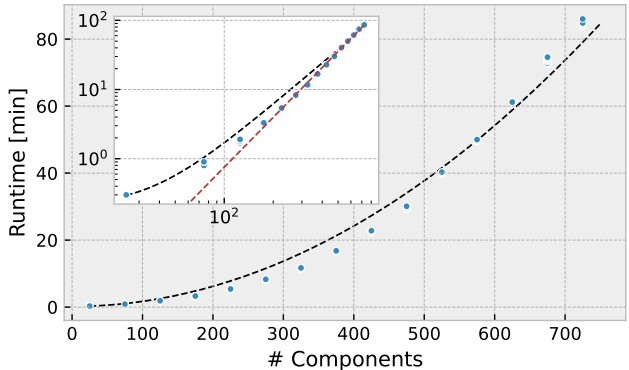

**Figure 4.** Computation time (wall time) for varying total network size. The total number of components is the sum of the numbers of different $P_i$, $H_i$, $D_i$, and $M_i$. For each size 16 different random communities were simulated for 100 years. Inset shows the same data in a log-log plot and a red dashed line with slope $\gamma = 2.4$.

## 3.2 Performance

MCoM is suited to simulate large communities comprising diverse species, DOM compounds, and metabolites. However, the numerical complexity and memory requirements of the simulation rise with system size. We assessed the performance of MCoM in a simple test recording run times for different system sizes. The results are shown in Figure 4. For each number $n \in \{5, 10, ..., 150\}$, we initialized 16 random networks with $n$ components of each type (heterotrophs, phototrophs, DOC, DON, and metabolites) and 30% connectivity. That is, for $n = 30$, each phototroph population produces 10 different DOC and 10 different DON compounds and is affected by 10 metabolites. Each heterotroph population produces the same variety of DOM compounds, is able to consume 10 DOC and 10 DON types, and produces 10 different metabolites. We simulated each community for 100 years with an integration step width of $dt = 0.025$ days, saving the last ten years of the simulation to disc saving the system's state every $dt_{\text{out}} = 0.5$ days. Asymptotically, the computation time scales as $n^\gamma$ with exponent $\gamma = 2.4$, as determined by the asymptotic slope of the graph of $\log n$ versus logarithmic computation time (dashed red line in the inset of Fig. 4). The black dashed curve shows a quadratic polynomial fit to the computation time.

## 4 Evaluation and application

### 4.1 Evaluation of network motifs without metabolite-induced feedbacks: Nutrient Remineralization in *Synechococcus* co-cultures

Here, we are evaluating model performance for individual motifs of the interaction network as the smallest unit in diverse microbial communities. In a first example, we apply the simplest version of the model, i.e., a co-culture of one phototrophic and one heterotrophic strain, without any metabolite-induced feedbacks, to an incubation experiment to assess its capability to reproduce the observed temporal variation in cell abundances. Christie-Oleza et al. (2017) cultured *Synechococcus* popu-





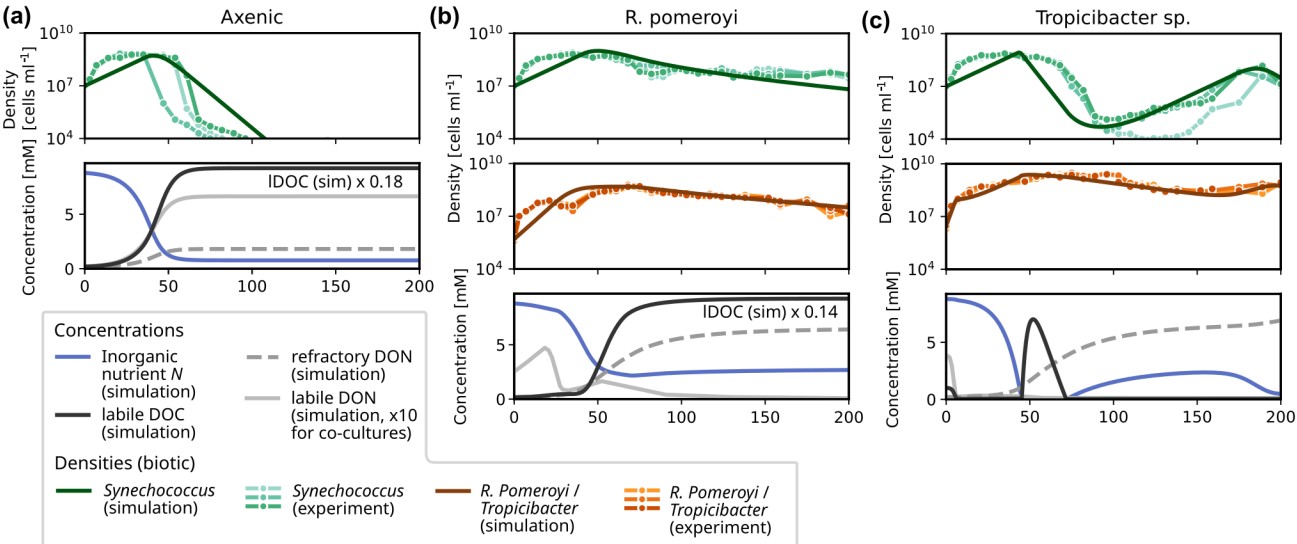

**Figure 5.** Experimentally observed and modelled growth dynamics of *Synechococcus* in co-culture with different heterotrophic bacteria: (a) axenic culture; (b) co-culture with *R. Pomeroyi*; (c) co-culture with *Tropicibacter sp.* For each experimental setup, three replicates were prepared, which are shown in the upper plot of each panel, see (Christie-Oleza et al., 2017) for details. For the simulations, we report *Synechococcus* (dark green) and heterotroph densities (brown), and concentrations of inorganic nutrient (blue), of labile DOC (black), labile DON (solid gray) and refractory DON (dashed gray).

lations over a 200-day period in co-cultures with two different heterotrophic bacteria (*R. Pomeroyi* and *Tropicibacter sp.*) in nutrient enriched ASW medium. When grown axenically, the phototroph population went extinct around day 75–100. If it was co-cultured with a heterotroph, it remained active until the end of the experiment at day 200. The decisive interaction was hypothesized to be an exchange of organic material, which provided the heterotrophs with energy and organically bound

nutrient, and of remineralized nutrient, which could be assimilated by *Synechococcus*. Although the phototroph population densities reached a peak around days 30–40 for each setup, they showed significantly distinct growth trajectories for the different types of heterotrophs. When grown together with *R. Pomeroyi*, the subsequent population decay proceeded steadily at a relatively slow rate until reaching $2.7 - 4.4 \times 10^7 \, \mathrm{cells \, ml^{-1}}$ at day 200. For *Tropicibacter sp.*, the temporal development after the initial peak is less stable (*cf.* Fig. 5c). Within the subsequent 60–70 days, *Synechococcus* collapsed to minima of

$1.0 - 13.0 \times 10^5 \, \mathrm{cells \, ml^{-1}}$ attained around days 96–117 for this case. These minima were then followed by a gradual recovery of until reaching another peak at density $3.6 - 14.7 \times 10^7 \, \mathrm{cells \, ml^{-1}}$ at the penultimate measurement time around day 190, declining again at the last measurement on day 200 to densities $1.1 - 2.1 \times 10^7 \, \mathrm{cells \, ml^{-1}}$.

We modelled both co-cultures and the axenic growth using MCoM with fixed growth parameters for the phototroph and specific characteristics for the heterotrophs. The exact parameters are listed in Table C1. Importantly, we did not assume

metabolite interactions, but the observations could be reproduced qualitatively by the exchange of nutrients in inorganic and organic form.





As an important characteristic of *R. Pomeroyi*, we incorporated the observation that it did not use the inorganic nutrient [as demonstrated in another experiment of Christie-Oleza et al. (2017)]. Further, we assumed that *R. Pomeroyi* immediately remineralizes 30% of DON during uptake, thus providing a steady nutrient source for *Synechococcus*. The simulation showed that the recycling of nutrients slowed down population decline significantly. However, to reproduce the steady decline observed in the experiment, a certain fraction of nutrients must continuously escape the recycling. Otherwise, the total nutrient in both populations and thus their densities would asymptote towards positive constants. We incorporated this by assuming that a fraction (10% for *R. Pomeroyi*) of the DON release is channelled into a refractory pool, which accumulates during the experiment.

For *Tropicibacter sp.*, we hypothesized that the more severe collapse of the *Synechococcus* population was partly caused by an ongoing competition for inorganic nutrient with *Tropicibacter sp.*, which is assumed to be a strong competitor (smaller $K_N$ than *Synechococcus*). Further, we modelled *Tropicibacter sp.* to behave more parsimoniously releasing only 5% of the DON it takes up in inorganic form. This assumption leads to a domination of the heterotroph until DOC is depleted and it becomes energy-limited, making inorganic nutrient available for *Synechococcus* once again and leading to the observed smaller second peak. Notably, the fit was significantly improved by assuming a more efficient and complete utilization of *Synechococcus* exudates by *Tropicibacter sp.* in comparison to *R. Pomeroyi*.

## 4.2 Evaluation of network motifs with metabolite-induced feedbacks: Interactions in *Prochlorococcus* co-cultures

In a next step, we evaluate MCoM's ability to model metabolite-induced feedbacks between heterotrophs and phototrophs. For this, we consider co-culture experiments of *Prochlorococcus* and different heterotrophic bacteria conducted by Sher et al. (2011), which were initially supplied with a stock of $0.8\,\mathrm{mM\,NH_4}$. In comparison to axenic cultures [Fig. 6(a)], Sher et al. observed different possible outcomes for different heterotrophic bacteria. Based on the cultures' observed bulk chlorophyll fluorescence over time, they clustered the heterotrophs into clearly separated groups exhibiting either inhibitory, neutral or growth-promoting effects on *Prochlorococcus* growth. Growth-promoting bacteria induced an earlier ($-4$ days on average) and more pronounced peak of the phototroph population than observed for the axenic (or neutrally affected) cases, while inhibitory bacteria significantly delayed the peak ($+13$ days on average).

We chose two representative examples from the experimental arrays published by Sher et al. (2011) for illustration. In co-culture with a *Rhodobacterales* strain (HOT5B8), Prochlorococcus growth was clearly promoted [Fig. 6(b)], whereas in co-culture with a *Marinobacter* strain (HOT4B5), growth was inhibited [Fig. 6(c)]. For the simulations, we used identical parameters for both heterotrophs and only varied the metabolite impact coefficient $a_M$ [*cf.* Eq. (9), simplified subscript] using $a_M = -0.2$ for the growth promoting case and $a_M = 0.17$ for the inhibitory case.

For both modelled co-cultures the simulated metabolite concentrations initially accumulate leading to a saturation at full effect strength around day 10 (purple curves in Fig. 6). For the growth-promoting case, metabolite effects decrease the loss rate of the *Prochlorococcus* population. Thereby the net growth rate of phototrophs is increased, leading to larger populations and accelerated depletion of the inorganic nutrient when compared to the axenic culture. For the inhibitory case, metabolite effects reduce the phytoplankton net growth rate. For both co-cultures, simulated DOC (black curves in Fig. 6) accumulates in the first days because the initial heterotrophic densities are low. This accumulation is less pronounced for the growth-





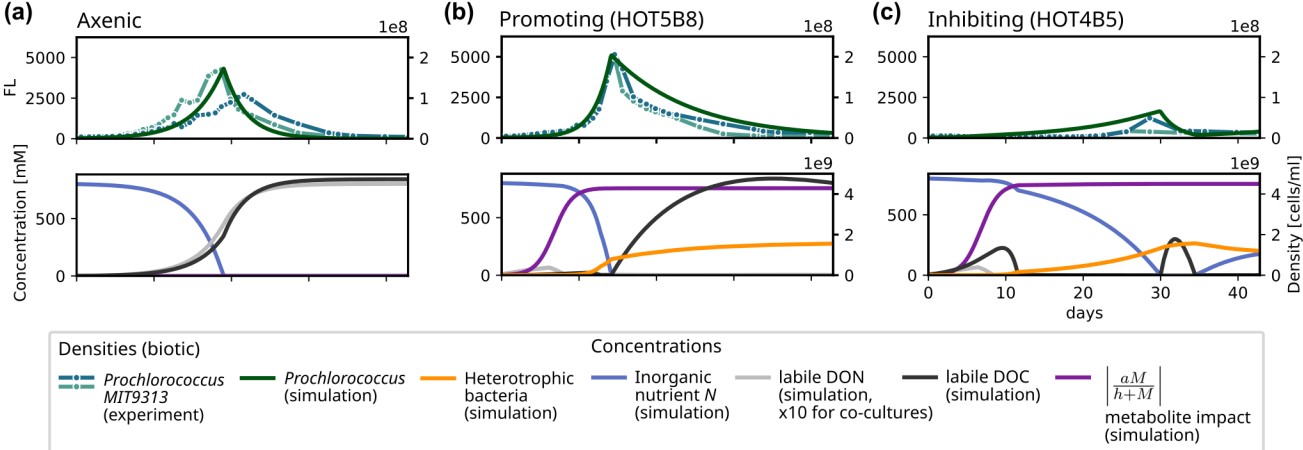

**Figure 6.** Experimentally observed and modelled growth dynamics of *Prochlorococcus* (MIT9313) in co-culture with different heterotrophic bacteria (Sher et al., 2011): (a) axenic culture; (b) co-culture with *Rhodobacterales* strain (HOT5B8); (c) co-culture with *Marinobacter* strain (HOT4B5). For each experimental setup, two replicates were prepared, which are shown in the upper plot of each panel. The simulated *Prochlorococcus* density (dark green curve) is plotted on top of the bulk chlorophyll fluorescence observed in the experiments. For the simulations, we report *Prochlorococcus* and heterotroph densities, $P$ and $H$, and concentrations of inorganic nutrient ($N$, blue), of labile DOC (black), and labile DON (solid gray). Further, we show the impact strength $\left| \frac{aM}{h+M} \right|$ of metabolites $M$ on the mortality rate of $P$ (solid purple). The model parameters used in this section are listed in Table C2.

promoting case due to the lower *Prochlorococcus* mortality. Around day 12 the accumulated DOC is depleted to minimal levels and heterotrophic growth becomes directly locked to DOC-release by phytoplankton. The DOC-locked regime lasts until the depletion of the inorganic nutrient stock, which coincides with the *Prochlorococcus* population peak. This happens at day 14 for the growth-promoting case, at day 19 for the axenic culture, and at day 30 for the inhibitory case. These differences

are caused by the different phytoplankton growth rates in the different scenarios, which largely determine the consumption rate of inorganic nutrient. After this point DOC starts to accumulate again, since heterotrophic growth becomes nutrient-limited. For the inhibitory case, this second phase of DOC accumulation is less pronounced due to the lower phytoplankton densities. Rather, we observe a second depletion of DOC at day 34 of the simulated experiment. This depletion is followed by a phase, where nutrient concentrations rise due to heterotrophic remineralization of surplus DON, comparable to the hypothesized

succession of events for the co-culture of *Synechococcus* and *Tropicibacter sp.* [Fig. 5(c)]. In conclusion, with a variation of single parameter ($a_M$) MCoM can capture both, the temporal shifts of the population peaks and the changes of the maximal observed *Prochlorococcus* density.





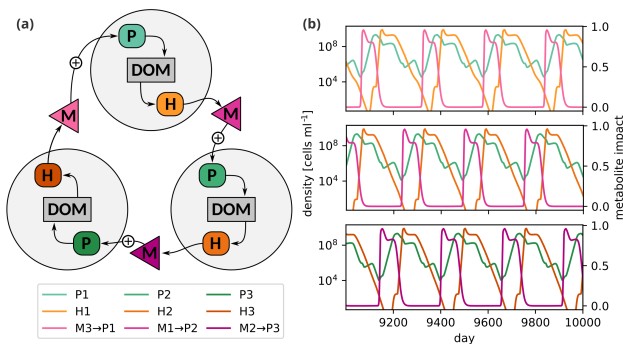

**Figure 7.** (a) Cyclic interaction scheme of three consortia; (b) Simulated population densities and metabolite impacts. The data of a single simulation is distributed over three panels to avoid cluttering of curves. See Table C3 for parameters.

## 4.3 Application for studying phytoplankton succession: Metabolite interactions lead to fluctuating dynamics

Having evaluated MCoMs ability to reproduce dynamics of individual motifs, which may represent components in a larger

microbial interaction network, we now show two examples of MCoM's application when scaled up to more diverse microbial communities. First, we consider a cyclic interaction network to illustrate how feedbacks mediated by metabolite interactions can lead to self-sustained fluctuations in a community of three phytoplankton and three heterotroph populations [Fig 7(a)]. We model associations of specific heterotroph and phytoplankton species, $H_i$ and $P_i$, by stipulating that the heterotroph $H_i$ specializes on the consumption of organic matter produced by the phototroph $P_i$, forming a "consortium". Due to this syntrophic

relationship population peaks of $P_i$ tend to be succeeded by peaks of $H_i$. Further, different consortia are assumed to be coupled via metabolites, such that metabolites produced by $H_i$ positively affect $P_{i+1}$. Effectively, this causes population peaks of $H_i$ to be succeeded by peaks of $P_{i+1}$, and so on, leading to a "merry-go-round" succession of consortia. Figure 7(b) shows the corresponding trajectory from day 9000 to 10000 after it has settled on a periodic orbit. While such a setup may appear highly artificial, it is robust to parameter variations and illustrates the potential of metabolite feedbacks to incite non-stationarity of

population densities even if no environmental forcing is present.[5]

## 4.4 Application for studying community stability: Metabolite interactions facilitate priority effects in microbial consortia

Generalizing the notion of a "consortium" as a coherent sub-community, we randomly generated communities consisting of separate highly connected groups of species. A consortium in this sense can loosely be defined as a set of phototrophs, DOM

compounds, heterotrophs and metabolites, which interact mostly within themselves. That is, within a consortium, heterotrophs feed on DOM released by members of the consortium, and phototroph growth is positively affected by metabolites produced by

---

[5]In natural situations, successions are regularly reported as a response to an initial environmental impulse, such as seasonal upwelling of nutrients. These scenarios can also be modelled with MCoM specifying a fluctuation for the parameter environment.nutrient.





**Figure 8.** Competition between two consortia for different inter-consortial coupling strength and initial states. (a) Schematic representation of the microbial community consisting of two consortia with inter-consortial coupling by shared metabolites. (b)–(d) Phytoplankton density trajectories; (b) and (c) Priority effects (bi-stability) in case of high connectivity and no coupling: The same interaction network may lead to a competitive exclusion of consortium B by consortium A (b) or *vice versa* (c), depending on which consortium is initially dominant; (d) Dynamics for higher inter-consortial metabolite interactions (overlap 5), leading to coexistence despite initial dominance of consortium A. (e) Periodic variation of the average environmental irradiance; (f) Asymptotic relative abundances for different coupling strengths and initial states. (g) Number of runs displaying coexistence.

these heterotrophs. Although competition between consortium members (for nutrients and DOM) is possible, positive feedback loops within a consortium can be expected to facilitate the growth of its members on average.

For each community, two consortia, A and B, were generated, each consisting of ten phototrophs, ten heterotrophs, ten DOC and ten DON compounds, and ten metabolite types. For the generation, we prescribed the in- and out-degrees of the microbial





nodes, i.e., the number of released and consumed compounds, as well as number of produced and effective metabolites (*cf.* Table C4). Then, we randomly connected the different components adhering to these degrees. Further, we allowed for a coupling of the two consortia by defining a number of shared metabolites, which effect both consortia. This number is called "overlap" in the following. Figure 8 (a) shows the community structure schematically.

In the following, we assumed an annual cycle of varying average light intensity as shown in Panel (e). Panels (b)–(d) show the time series of simulated phytoplankton population densities for different degrees of inter-consortial coupling and different initial conditions. In Panel (b) and (c), we show two simulations for the same community consisting of two consortia with a single overlapping metabolite. This configuration often leads to priority effects, as exemplified by the depicted dynamics: When the phototrophs of consortium A (green curves) are initialized with higher density than the phototrophs of consortium

B [see Panel (b)], consortium A remains dominant over the whole time span of the simulation (approx. 45 years). *Vice versa*, consortium B remains dominant if its phototrophs have higher initial density [Panel (c)].

Panel (d) shows a trajectory in a modified network with higher inter-consortial coupling (overlap 5), where priority effects are not observed, i.e., despite differing initial phototroph densities the trajectories converge to identical periodic orbits after a transient time. In such cases, the 'crosstalk' between consortia prevents their dynamical separation and an equilibrium, where

members of both consortia coexist, is attained independently of the initial state.

We explored this effect systematically in different systems. For each of six values of the overlap, we generated 20 communities, each containing two consortia, A and B (20 phototrophs and 20 heterotrophs in total). Each community was initialized twice, with either consortium A or consortium B being dominant, and integrated for 60 years (see Table C4 for a detailed description of parameters). The relative averaged abundance over the last ten years of the simulation is reported for each simu-

lation run in Panel (f). Starting from different initial states, where either consortium A or B is dominant, priority effects appear as differences in observed asymptotic distributions for the different initial states.

We categorized all observed regimes either as coexistence, where both consortia contribute more than 10% to the total abundance, or as dominance of either consortium A or B. For low overlap, communities show pronounced priority effects in most cases, where in general the initially dominant consortium remains dominant. For instance, for zero overlap, none of

the simulations ended in coexistence [Panel (g)]. However, in 1/8 of the simulations the finally dominating consortium was the initially rare one. For one overlapping metabolite, two of 40 runs led to a coexistence of both consortia. Already for an overlap of three metabolites about 50% of the simulations end in coexistence, and for overlap higher than five (where >30% of metabolites are shared between consortia), species coexistence is the most frequent (>75%) simulation outcome.

## 5    Conclusions

MCoM v1.0 is a versatile, scalable framework for simulating the dynamics of microbial communities consisting of phototrophic and heterotrophic species that includes a wide range of microbial interactions. The processes implemented into MCoM capture essential mechanisms of these interactions, such as nutrient competition, exo-metabolite and DOM production, as well as remineralization. Due to its flexible structure, MCoM allows to explore a range of ecological scenarios, from single-species





experiments (Sher et al., 2011; Christie-Oleza et al., 2017) to complex community interactions (Garcia et al., 2018; Kost et al., 2023). We demonstrated this by simulating simple co-culture experiments, as well as non-linear phenomena such as emergent periodic succession patterns and multi-stability, which are prerequisites for modelling, e.g., ecological tipping points. Aside from aiding mechanistic insights to ecological observations, MCoM may prove useful for the simulation of biotechnological setups as the relevance and the potential of microbial interactions for industrial exploitation becomes increasingly recognized (Ramanan et al., 2016).

MCoM is intentionally kept relatively simple in order to assess fundamental controls of microbial communities and their impact on biogeochemical fluxes. Due to its customizable nature, it offers a robust foundation for future enhancements. Several more detailed process descriptions could be integrated as modular additions, which can be toggled on or off according to the user needs. For instance, incorporating adaptive elemental ratios that vary with environmental conditions (Godwin and Cotner, 2015; Trautwein et al., 2017; Tanioka and Matsumoto, 2020) could further refine its representation of both phytoplankton and heterotrophic processes. Introducing adaptive interaction strategies, as seen in algicidal interactions (Seyedsayamdost et al., 2011; Meyer et al., 2017) and expanding the role of exo-metabolites to influence assimilation rates or other parameters (beyond phytoplankton mortality) would increase the model's versatility. Furthermore, incorporating environmental factors like temperature to modulate growth rates (Coles and Jones, 2000; Bouman et al., 2005) could improve the model's accuracy in scenarios with varying environments. In summary, MCoM provides a versatile platform that is customizable to specific requirements from assessing pairwise interactions to diverse microbial communities. By taking into account the full range of positive and negative interactions, it expands the currently prevailing competition-centred view in biogeochemical modelling.

*Code availability.* The source code for MCoM v1.0 is published under MIT license and can be downloaded from Zenodo (Lücken et al., 2025). The repository contains the software documentation, instructions on configuration and installation, as well as a test suite. To run MCoM, Julia and Python must be installed (see README.md and requirements.txt for more specific dependencies). For the latest version, please visit https://github.com/bgc-mod/MCoM.

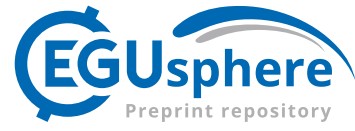

# Appendix A: Reference tables





**Table A1.** Parameters of the MCoM model. All parameters are assumed to be non-negative if not stated otherwise.

| Parameter | Description | Source code variable | Unit |
|---|---|---|---|
| $V_{i,N}$ | Maximal DIN uptake rates of $P_i$ or $H_i$ | V_NP[i], V_NH[i] | $\mathrm{fmol(N)\,day^{-1}\,cell^{-1}}$ |
| $K_{i,N}$ | Half saturation constants for DIN uptake by $P_i$ or $H_i$ | K_NP[i], K_NH[i] | $\mathrm{mM(N)}$ |
| $V_{i,j}$ | Maximal uptake rates DOC (resp. DON) compound $j$ uptake by $H_i$ | V_DOCH[j,i], V_DONH[j,i] | $\mathrm{fmol(C)\,day^{-1}\,cell^{-1}}$, $\mathrm{fmol(N)\,day^{-1}\,cell^{-1}}$ |
| $K_{i,j}$ | Half saturation constants for DOC (resp. DON) compound $j$ uptake by $H_i$ | K_DOCH[j,i], K_DONH[j,i] | $\mathrm{mM(C)}$, $\mathrm{mM(N)}$ |
| $Y_{i,j}$ | Biomass yield coefficients, i.e., fractions of uptake of DOC (resp. DON) compound $D_j$ integrated into biomass of $P_i$ or $H_i$, $0 \leq Y_{i,j} \leq 1$ | Y_DOC[j,i], Y_DON[j,i] | - |
| $\chi_i^C$ | Carbon content per cell of population $P_i$ or $H_i$ | XH_C[i], XP_C[i] | $\mathrm{fmol(C)\,cell^{-1}}$ |
| $r_i^{C:N}, r_{\mathrm{ex},i}^{C:N}$ | C:N ratios of populations $P_i$ or $H_i$, and of $N$-rich exudates of $P_i$, $r_i^{C:N} > r_{\mathrm{ex},i}^{C:N}$ for all species. | rCN_P[i], rCN_H[i] | $\mathrm{mol(C)\,mol(N)^{-1}}$ |
| $r_i^{\mathrm{Chl}:C}$ | Chlorophyll to carbon ratio for cells of $P_i$ | rChlC[i] | $\mathrm{mol(Chl)\,mol(C)^{-1}}$ |
| $\chi_i^N, \chi_i^{\mathrm{Chl}}$ | Nutrient or chlorophyll content per cell ($\chi_i^N = \chi_i^C / r_i^{C:N}$, $\chi_i^{\mathrm{Chl}} = \chi_i^C \cdot r_i^{Chl:C}$) | - | $\mathrm{fmol(N)\,cell^{-1}}$, $\mathrm{fmol(Chl)\,cell^{-1}}$ |
| $q_i^{\mathrm{ex}}$ | Maximal fractions of assimilated DOC used for DON exudation of $P_i$ in light limited regime. | q_ex[i] | - |
| $\delta_i$ | Linear loss rates of phytoplankton, heterotrophs, and metabolites | d_P[i], d_H[i], d_M[i] | $\mathrm{day^{-1}}$ |
| $\delta_{q,i}$ | Quadratic loss rates of phytoplankton and heterotrophs | d_P2[i]*XP_C[i], d_H2[i]*XH_C[i] | $\mathrm{cell^{-1}\,day^{-1}}$ |
| $a_{i,j}^M$ | Interaction rate per unit of metabolite $j$ on growth of phytoplankton population $i$; can be negative | lambda_i *A_MP[j,i] | $\mathrm{day^{-1}}$ |
| $h_{i,j}$ | Half saturation constant for the effect of metabolite $j$ on population $i$ | h_M[j,i] | - |
| $\theta_i$ | Conversion coefficient for determining the amount of metabolite $i$ corresponding to the amount of invested DOC | not included (assumed equal to one) | $\mathrm{mM(C)^{-1}}$ |
| $\alpha_i$ | Slope of the P-I curve of $P_i$ at irradiance $I = 0$. (Effective if variant.use_PI_curve=true) | I_alpha[i] | $\frac{\mathrm{mol(C)\,m^2}}{86400\,\mathrm{mol(Chl)\,\mu mol(Q)}}$ |
| $\beta_i$ | Photoinhibition coefficient of the P-I curve of $P_i$. (Effective if variant.use_PI_curve=true) | I_beta[i] | $\frac{\mathrm{mol(C)\,m^2}}{86400\,\mathrm{mol(Chl)\,\mu mol(Q)}}$ |
| $\phi_{s,i}$ | Upper limit for the photosynthesis rate of $P_i$. (Effective if variant.use_PI_curve=true) | I_max[i] | $\mathrm{mol(C)\,mol(Chl)^{-1}\,day^{-1}}$ |
| $R_{j,i}$ | Release partition coefficients, $0 \leq R_{j,i} \leq 1$, for $i \in \mathcal{P}$ or $i \in \mathcal{H}$, and $j \in \mathcal{C}$ or $j \in \mathcal{N}$ | R_PDOC[i,j], R_PDON[i,j], R_HDOC[i,j], R_HDON[i,j] | - |
| $\pi_{j,i}$ | Production of metabolite $j$ by heterotroph $H_i$ | P_HM[i,j] | - |
| irrad_min, irrad_max | Range of the average irradiance. (Varies sinusoidally between minimum and maximum in one year.) If variant.use_PI_curve=false, this is interpreted directly as specific photosynthesis rate with unit $\frac{\mathrm{\mu mol\,C}}{\mathrm{\mu mol\,Chl \cdot day}}$. | irrad_min, irrad_max | $\mathrm{\mu mol(Q)\,m^{-2}\,s^{-1}}$ |



**Table A2.** Reference table of notations for elemental flows.

| Flow | Description | Source | Target | Unit |
|---|---|---|---|---|
| **General population-associated flows** ($i \in \mathcal{P}$ or $i \in \mathcal{H}$) | | | | |
| $f_{N \to i}$ | realized inorganic nutrient assimilation into biomass | DIN | $P_i$ or $H_i$ | $\mathrm{fmol(N)\,day^{-1}}$ |
| $f_{N \to i}^{\max}$ | maximal inorganic nutrient assimilation into biomass | DIN | $P_i$ or $H_i$ | $\mathrm{fmol(N)\,day^{-1}}$ |
| $f_{i \to DOC}$ | total biomass loss to DOC | $P_i$ or $H_i$ | DOC | $\mathrm{fmol(C)\,day^{-1}}$ |
| $f_{i \to DON}$ | total biomass loss to DON | $P_i$ or $H_i$ | DON | $\mathrm{fmol(N)\,day^{-1}}$ |
| **Phytoplankton populations** ($i \in \mathcal{P}$) | | | | |
| $f_{C \to i}$ | realized photosynthetic carbon assimilation into biomass | DIC | $P_i$ | $\mathrm{fmol(C)\,day^{-1}}$ |
| $f_{C \to i}^{\max}$ | photosynthetic carbon fixation (= maximal assimilation) | DIC | $P_i$ | $\mathrm{fmol(C)\,day^{-1}}$ |
| $f_{C \to DOC}^{i}$ | DOC exudation | DIC | DOC | $\mathrm{fmol(C)\,day^{-1}}$ |
| $f_{N \to DON}^{i}$ | DON exudation | DIN | DON | $\mathrm{fmol(N)\,day^{-1}}$ |
| **Heterotroph populations** ($i \in \mathcal{H}$) | | | | |
| $f_{DOC \to i}, f_{DON \to i}$ | realized total organic carbon, resp. nutrient, assimilation into biomass | DOC, DON | $H_i$ | $\mathrm{fmol(C)\,day^{-1}}, \mathrm{fmol(N)\,day^{-1}}$ |
| $f_{DOC \to i}^{\max}, f_{DON \to i}^{\max}$ | maximal total organic carbon, resp. nutrient, assimilation into biomass | DOC, DON | $H_i$ | $\mathrm{fmol(C)\,day^{-1}}, \mathrm{fmol(N)\,day^{-1}}$ |
| $f_{j \to i}^{up}, j \in \mathcal{C} \cup \mathcal{N}$ | realized uptake of DOC or DON compound $j$ into biomass | $D_j$ | $H_i$, and DIC or DIN | $\mathrm{fmol(C)\,day^{-1}}, \mathrm{fmol(N)\,day^{-1}}$ |
| $f_{j \to i}^{up,\max}, j \in \mathcal{C} \cup \mathcal{N}$ | maximal uptake of DOC or DON compound $j$ into biomass | $D_j$ | $H_i$, and DIC or DIN | $\mathrm{fmol(C)\,day^{-1}}, \mathrm{fmol(N)\,day^{-1}}$ |
| $f_{j \to C}^{i}, f_{j \to N}^{i}, j \in \mathcal{C} \cup \mathcal{N}$ | remineralization of DOC or DON compound $j$ | $D_j$ | DIC, DIN | $\mathrm{fmol(C)\,day^{-1}}, \mathrm{fmol(N)\,day^{-1}}$ |
| $f_{DON \to N}^{+}$ | surplus nutrient remineralization | DON | DIN | $\mathrm{fmol(N)\,day^{-1}}$ |
| **DOC or DON compounds** ($j \in \mathcal{C}$ or $j \in \mathcal{N}$) | | | | |
| $f_{H \to j}$ | total heterotroph biomass loss to compound $j$ | $H_i$ | $D_j$ | $\mathrm{fmol(C)\,day^{-1}}, \mathrm{fmol(N)\,day^{-1}}$ |
| $f_{P \to j}$ | total phototroph biomass loss to compound $j$ | $P_i$ | $D_j$ | $\mathrm{fmol(C)\,day^{-1}}, \mathrm{fmol(N)\,day^{-1}}$ |
| $f_{C \to j}, f_{N \to j}$ | total phototrophic exudation of compound $j$ | DIN, DOC | $D_j$ | $\mathrm{fmol(C)\,day^{-1}}, \mathrm{fmol(N)\,day^{-1}}$ |
| $f_{C \to j}^{i}, f_{N \to j}^{i}, i \in \mathcal{P}$ | exudation of compound $j$ by population $P_i$ | DIN, DOC | $D_j$ | $\mathrm{fmol(C)\,day^{-1}}, \mathrm{fmol(N)\,day^{-1}}$ |
| $f_{j \to H}$ | total assimilation of compound $j$ into heterotroph biomass | $D_j$ | $H_i$ | $\mathrm{fmol(C)\,day^{-1}}, \mathrm{fmol(N)\,day^{-1}}$ |
| $f_{j \to C}, f_{j \to N}$ | total remineralization of compound $j$ | $D_j$ | DIC, DIN | $\mathrm{fmol(C)\,day^{-1}}, \mathrm{fmol(N)\,day^{-1}}$ |
| $f_{i \to j}, i \in \mathcal{P} \cup \mathcal{H}$ | biomass loss of population $i$ to compound $j$ | $P_i$ or $H_i$ | $D_j$ | $\mathrm{fmol(C)\,day^{-1}}, \mathrm{fmol(N)\,day^{-1}}$ |
| **Nutrient $N$** | | | | |
| $f_{DON \to N}$ | total remineralization | DON | DIN | $\mathrm{fmol(N)\,day^{-1}}$ |
| $f_{N \to P}, f_{N \to H}$ | total assimilation into biomass | DIN | $P_i$ or $H_i$ | $\mathrm{fmol(N)\,day^{-1}}$ |





## Appendix B: Calculation of realized carbon assimilation ($f_{C \rightarrow i}$) and DOC exudation rates ($f^i_{C \rightarrow \mathrm{DOC}}$)

We use the stoichiometric ratios $r^{C:N}_i$ and $r^{C:N}_{\mathrm{ex},i}$, the maximal assimilation rates $f^{\max}_{C \rightarrow i}$ and $f^{\max}_{N \rightarrow i}$, and the maximal fraction $q^{\mathrm{ex}}_i$ of carbon allocated for exudation to derive the realized assimilation and exudation rates $f_{C \rightarrow i}$ and $f^i_{C \rightarrow \mathrm{DOC}}$.

We assume that carbon is fixated at maximum rate. Since it is fully allocated for assimilation and exudation [Eq. (6)], we have

$$f_{C \rightarrow i} = f^{\max}_{C \rightarrow i} - f^i_{C \rightarrow \mathrm{DOC}}. \tag{B1}$$

Further, the required uptake of $N$ by $P_i$, that is $f^i_{N \rightarrow \mathrm{DON}} + f_{N \rightarrow i}$, is related to the total carbon fixation by the stoichiometric composition of assimilation and exudation fluxes [Eqs. (2) and (7)]. Dividing these by the ratios, adding them up, then using

Eq. (B1) and some basic algebra gives

$$f_{N \rightarrow i} + f^i_{N \rightarrow \mathrm{DON}} = \frac{r^{C:N}_{\mathrm{ex},i}}{r^{C:N}_i r^{C:N}_{\mathrm{ex},i}} \cdot f^{\max}_{C \rightarrow i} + \frac{r^{C:N}_i - r^{C:N}_{\mathrm{ex},i}}{r^{C:N}_i r^{C:N}_{\mathrm{ex},i}} \cdot f^i_{C \rightarrow \mathrm{DOC}}. \tag{B2}$$

Note that the coefficient of $f^i_{C \rightarrow \mathrm{DOC}}$ is positive as we assume $r^{C:N}_i > r^{C:N}_{\mathrm{ex},i}$.

This can be solved for $f^i_{C \rightarrow \mathrm{DOC}}$ by maximization under the given constraints

$$f^i_{C \rightarrow \mathrm{DOC}} \leq q^{\max}_i f^{\max}_{C \rightarrow i}, \tag{B3}$$

and

$$f_{N \rightarrow i} + f^i_{N \rightarrow \mathrm{DON}} \leq f^{\max}_{N \rightarrow i},$$

which, using Eq. (B2), is equivalent to

$$f^i_{C \rightarrow \mathrm{DOC}} \leq \frac{r^{C:N}_{\mathrm{ex},i}}{r^{C:N}_i - r^{C:N}_{\mathrm{ex},i}} \left( r^{C:N}_i f^{\max}_{N \rightarrow i} - f^{\max}_{C \rightarrow i} \right). \tag{B4}$$

Hence,

$$f^i_{C \rightarrow \mathrm{DOC}} = \min \left( q^{\max}_i f^{\max}_{C \rightarrow i}, \frac{r^{C:N}_{\mathrm{ex},i}}{r^{C:N}_i - r^{C:N}_{\mathrm{ex},i}} \left( r^{C:N}_i f^{\max}_{N \rightarrow i} - f^{\max}_{C \rightarrow i} \right) \right).$$



## Appendix C: Parameters used in Section 4

**Table C1.** Parameters for modeling Co-culture experiments of *Synechococcus* and heterotrophic bacteria, see Sec. 4.1.

| Parameter | *Synechococcus* | Parameter | *R. Pomeroyi* | *Tropicibacter sp.* |
|---|---|---|---|---|
| $P_0$ | $10^7\,\mathrm{cells\,ml^{-1}}$ | $H_0$ | $5 \cdot 10^5\,\mathrm{cells\,ml^{-1}}$ | $3 \cdot 10^6\,\mathrm{cells\,ml^{-1}}$ |
| $\chi_C$ | $12.0\,\mathrm{fmol(C)\,cell^{-1}}$ | $\chi_C$ | $15.0\,\mathrm{fmol(C)\,cell^{-1}}$ | $12.0\,\mathrm{fmol(C)\,cell^{-1}}$ |
| $r_i^{C:N}$ | $5.2\,\mathrm{mol(C)\,mol(N)^{-1}}$ | $r_i^{C:N}$ | $4.0\,\mathrm{mol(C)\,mol(N)^{-1}}$ | $4.0\,\mathrm{mol(C)\,mol(N)^{-1}}$ |
| $f_{C \to i}^{\max}$ | $4.6\,\mathrm{fmol(C)\,day^{-1}\,cell^{-1}}$ (constant photosynthetic rate) | $\delta, \delta_q$ | $0.1\,\mathrm{day^{-1}},\,0.45\,(10^6\,\mathrm{cells})^{-1}\,\mathrm{day^{-1}}$ | $0.2\,\mathrm{day^{-1}},\,0.012\,(10^6\,\mathrm{cells})^{-1}\,\mathrm{day^{-1}}$ |
| $\delta, \delta_q$ | $0.28\,\mathrm{day^{-1}},\,0.0\,\mathrm{cell^{-1}\,day^{-1}}$ | $V_N$ | $0.0\,\mathrm{fmol(N)\,cell^{-1}\,day^{-1}}$ | $4.0\,\mathrm{fmol(N)\,cell^{-1}\,day^{-1}}$ |
| $V_N$ | $1.7\,\mathrm{fmol(N)\,cell^{-1}\,day^{-1}}$ | $K_N$ | - | $0.1\,\mathrm{mM(N)}$ |
| $K_N$ | $2.0\,\mathrm{mM(N)}$ | $V_{DON}$ | $4.0\,\mathrm{fmol(N)\,cell^{-1}\,day^{-1}}$ | $5.2\,\mathrm{fmol(N)\,cell^{-1}\,day^{-1}}$ |
| | | $K_{DON}$ | $0.1\,\mathrm{mM(N)}$ | $0.1\,\mathrm{mM(N)}$ |
| | **Environment** | $Y_{DON}$ | $0.7$ | $0.95$ |
| $N_0$ | $8.8\,\mathrm{mM(N)}$ | $V_{DOC}$ | $6.0\,\mathrm{fmol(N)\,cell^{-1}\,day^{-1}}$ | $12.2\,\mathrm{fmol(N)\,cell^{-1}\,day^{-1}}$ |
| $\Delta$ | $0.0\,\mathrm{day^{-1}}$ | $K_{DOC}$ | $0.1\,\mathrm{mM(C)}$ | $0.1\,\mathrm{mM(C)}$ |
| | | $Y_{DOC}$ | $0.8$ | $0.87$ |
| | | $R_{lDOC,H}$ | $0.5$ | $0.99$ |
| | | $R_{lDOC,P}$ | $0.9$ | $1.0$ |
| | | $R_{lDON,H}$ | $0.9$ | $0.93$ |
| | | $R_{lDON,P}$ | $0.9$ | $1.0$ |





**Table C2.** Parameters for modeling Co-culture experiments of *Prochlorococcus* and heterotrophic bacteria, see Sec. 4.2.

| | *Prochlorococcus* | | **Heterotrophs** | |
|---|---|---|---|---|
| $P_0$ | $10^6 \, \mathrm{cells \, ml^{-1}}$ | $H_0$ | $10^4 \, \mathrm{cells \, ml^{-1}}$ |
| $\chi_C$ | $10.0 \, \mathrm{fmol(C) \, cell^{-1}}$ | $\chi_C$ | $2.0 \, \mathrm{fmol(C) \, cell^{-1}}$ |
| $r_i^{C:N}$ | $5.0 \, \mathrm{mol(C) \, mol(N)^{-1}}$ | $r_i^{C:N}$ | $4.0 \, \mathrm{mol(C) \, mol(N)^{-1}}$ |
| $f_{C \to i}^{\max}$ | $10.0 \, \mathrm{fmol(C) \, day^{-1} \, cell^{-1}}$ (constant photosynthetic rate) | $\delta, \delta_q$ | $0.2 \, \mathrm{day^{-1}}, \, 0.0 \, \mathrm{cell^{-1} \, day^{-1}}$ |
| $\delta, \delta_q$ | $0.35 \, \mathrm{day^{-1}}, \, 0.0 \, \mathrm{cell^{-1} \, day^{-1}}$ | $V_N$ | $1.5 \, \mathrm{fmol(N) \, cell^{-1} \, day^{-1}}$ |
| $V_N$ | $1.25 \, \mathrm{fmol(N) \, cell^{-1} \, day^{-1}}$ | $K_N$ | $8.0 \, \mathrm{mM(N)}$ |
| $K_N$ | $2.4 \, \mathrm{mM}$ | $V_{DON}$ | $3.0 \, \mathrm{fmol(N) \, cell^{-1} \, day^{-1}}$ |
| | | $K_{DON}$ | $0.6 \, \mathrm{mM(N)}$ |
| | **Environment** | $Y_{DON}$ | $0.6$ |
| $N_0$ | $800 \, \mathrm{mM(N)}$ | $V_{DOC}$ | $3.0 \, \mathrm{fmol(N) \, cell^{-1} \, day^{-1}}$ |
| $\Delta$ | $0.0 \, \mathrm{day^{-1}}$ | $K_{DOC}$ | $0.6 \, \mathrm{mM(C)}$ |
| $l\mathrm{DOC}_0$ | $5.0 \, \mathrm{mM(C)}$ | $Y_{DOC}$ | $0.7$ |
| $l\mathrm{DON}_0$ | $0.05 \, \mathrm{mM(N)}$ | $h_M$ | $5 \cdot 10^{-7}$ |
| $M_0$ | $10^{-9}$ | $\pi_M$ | $10^{-7}$ |
| $\delta_M$ | $0.3 \, \mathrm{day^{-1}}$ | | |
| | *Rhodobacterales* **(HOT5B8)** | | *Marinobacter* **(HOT4B5)** |
| $a_M$ | $-0.2$ | $a_M$ | $0.17$ |



**Table C3.** Parameters used for the cyclic interaction motif, see Sec. 4.3.

| Parameter | Phytoplankton | Parameter | Heterotrophs |
|---|---|---|---|
| $\chi_i^C$ | $10.0\,\text{fmol(C)}\,\text{cell}^{-1}$ | $\chi_i^C$ | $2.0\,\text{fmol(C)}\,\text{cell}^{-1}$ |
| $r_i^{C:N}$ | $5.2\,\text{mol(C)}\,\text{mol(N)}^{-1}$ | $r_i^{C:N}$ | $4.0\,\text{mol(C)}\,\text{mol(N)}^{-1}$ |
| $\delta_i, \delta_{q,i}$ | $0.2\,\text{day}^{-1}, 0.2\,(10^6\,\text{cells})^{-1}\,\text{day}^{-1}$ | $\delta_i, \delta_{q,i}$ | $0.1\,\text{day}^{-1}, 0.02\,(10^6\,\text{cells})^{-1}\,\text{day}^{-1}$ |
| $h_{i,j}$ | $10^{-4}$ | $\pi_{j,i}$ | $10^{-5}$ |
| $a_{i,j}^M$ | $0.0$ or $-1.0$ | **Nutrient uptake** | |
| $V_{i,N}$ | $1.0\,\text{fmol(N)}\,\text{cell}^{-1}\,\text{day}^{-1}$ | $V_{i,N}$ | $1.5\,\text{fmol(N)}\,\text{cell}^{-1}\,\text{day}^{-1}$ |
| $K_{i,N}$ | $2.0\,\text{mM(N)}$ | $K_{i,N}$ | $8.0\,\text{mM(N)}$ |
| $\phi_{s,i}$ | $5.0\,\text{mol(C)}\,\text{mol(Chl)}^{-1}\,\text{day}^{-1}$ | **DON uptake** | |
| $\alpha_i$ | $0.08\,\frac{\text{mol(C)}\,\text{m}^2}{86400\,\text{mol(Chl)}\,\mu\text{mol(Q)}}$ | $V_{i,j}$ | $3.0\,\text{fmol(N)}\,\text{cell}^{-1}\,\text{day}^{-1}$ |
| $\beta_i$ | $0.003\,\frac{\text{mol(C)}\,\text{m}^2}{86400\,\text{mol(Chl)}\,\mu\text{mol(Q)}}$ | $K_{i,j}$ | $1.0\,\text{mM(N)}$ |
| | | $Y_{i,j}$ | $0.5$ |
| **Environment** | | **DOC uptake** | |
| $\Delta$ | $0.1\,\text{day}^{-1}$ | $V_{i,j}$ | $4.0\,\text{fmol(N)}\,\text{cell}^{-1}\,\text{day}^{-1}$ |
| $\delta_M$ | $0.1\,\text{day}^{-1}$ | $K_{i,j}$ | $1.0\,\text{mM(C)}$ |
| $N_{\text{ext}}$ | $5.0\,\text{mM(N)}$ | $Y_{i,j}$ | $0.5$ |
| $I$ | $10.0\,\mu\text{mol(Q)}\,\text{m}^{-2}\,\text{day}^{-1}$ | | |



**Table C4.** Parameters used for the simulation of two competing consortia, see Sec. 4.4

| Parameter | Phytoplankton | Parameter | Heterotrophs |
|---|---|---|---|
| $P_i(0)$ | $10^8$, resp. $10^6$, cells ml$^{-1}$ | $H_i(0)$ | $10^5$–$10^7$ cells ml$^{-1}$ |
| $\chi_i^C$ | $10.0\,\mathrm{fmol(C)\,cell^{-1}}$ | $\chi_i^C$ | $2.0\,\mathrm{fmol(C)\,cell^{-1}}$ |
| $r_i^{C:N}$ | $5.2\,\mathrm{mol(C)\,mol(N)^{-1}}$ | $r_i^{C:N}$ | $4.0\,\mathrm{mol(C)\,mol(N)^{-1}}$ |
| $\delta_i, \delta_{q,i}$ | $0.1\,\mathrm{day^{-1}}, 1.0\,(10^6\,\mathrm{cells})^{-1}\,\mathrm{day^{-1}}$ | $\delta_i, \delta_{q,i}$ | $0.1\,\mathrm{day^{-1}}, 0.02\,(10^6\,\mathrm{cells})^{-1}\,\mathrm{day^{-1}}$ |
| $h_{i,j}$ | $10^{-4}$ | $\pi_{j,i}$ | $10^{-5}$ |
| $a_{i,j}^M$ | $0.0$ or $-0.1$ | **Nutrient uptake** | |
| $V_{i,N}$ | $1.0\,\mathrm{fmol(N)\,cell^{-1}\,day^{-1}}$ | $V_{i,N}$ | $1.5\,\mathrm{fmol(N)\,cell^{-1}\,day^{-1}}$ |
| $K_{i,N}$ | $2.0\,\mathrm{mM(N)}$ | $K_{i,N}$ | $8.0\,\mathrm{mM(N)}$ |
| $\phi_{s,i}$ | $3.0$–$5.0\,\mathrm{mol(C)\,mol(Chl)^{-1}\,day^{-1}}$ | **DON uptake** | |
| $\alpha_i$ | $0.05$–$0.3\ \frac{\mathrm{mol(C)\,m^2}}{86400\,\mathrm{mol(Chl)\,\mu mol(Q)}}$ | $V_{i,j}$ | $3.0\,\mathrm{fmol(N)\,cell^{-1}\,day^{-1}}$ |
| $\beta_i$ | $0.003$–$0.08\ \frac{\mathrm{mol(C)\,m^2}}{86400\,\mathrm{mol(Chl)\,\mu mol(Q)}}$ | $K_{i,j}$ | $1.0\,\mathrm{mM(N)}$ |
| | | $Y_{i,j}$ | $0.5$ |
| **Environment** | | **DOC uptake** | |
| $\Delta$ | $0.1\,\mathrm{day^{-1}}$ | $V_{i,j}$ | $4.0\,\mathrm{fmol(N)\,cell^{-1}\,day^{-1}}$ |
| $N_0$ | $8.8\,\mathrm{mM(N)}$ | $K_{i,j}$ | $1.0\,\mathrm{mM(C)}$ |
| $\delta_M$ | $0.1\,\mathrm{day^{-1}}$ | $Y_{i,j}$ | $0.5$ |
| $N_{\mathrm{ext}}$ | $5.0\,\mathrm{mM(N)}$ | | |
| $I_{\mathrm{min}}$ | $0.0\,\mathrm{\mu mol(Q)\,m^{-2}\,day^{-1}}$ | **Connectivity** | |
| $I_{\mathrm{max}}$ | $125.0\,\mathrm{\mu mol(Q)\,m^{-2}\,day^{-1}}$ | exPDOC | 4 |
| | | exPDON | 4 |
| **System dimensions** | | exHDOC | 4 |
| dims.P | 20 | exHDON | 4 |
| dims.H | 20 | upDOCH | 3 |
| dims.DOC | 20 | upDONH | 3 |
| dims.DON | 20 | prodHM | 2 |
| dims.M | 10–20 (overlap-depd.) | effsMP | 4 |



*Author contributions.* All authors added to the conceptualization of the model; J.G.B., M.J.F. and S.T.L. contributed to an initial implementation of the model; L.L. transferred and extended the code into its current form. S.T.L. and L.L. conducted the simulation experiments; L.L. wrote the initial manuscript draft; All authors reviewed the manuscript.

*Competing interests.* The authors declare that they have no conflict of interest.

*Acknowledgements.* We are grateful for support from the Simons Foundation, CBIOMES Award 549931 to M.J.F. S.T.L. acknowledges funding from German Research Foundation (DFG), grant no.: 445120363, as well as from the Ministerium für Wissenschaft und Kultur Niedersachsen (MWK, Grant 16TTP079). L.L. and S.T.L. acknowledge funding from the Simons Foundation, Award 01060273. We thank Joseph Christie de Oleza and Daniel Sher for sharing their experimental data with us.



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
