# Peer review of "The microbial community model MCoM 1.0: A scalable framework for modelling phototroph-heterotrophic interactions in diverse microbial communities"

_EGUsphere, 2025_

## Referee Comment (RC1)

**Review of "*The microbial community model MCoM 1.0: A scalable framework for modelling phototroph-heterotrophic interactions in diverse microbial communities*" by Lücken, Follows, Bragg and Lennart**

**General comments**

This paper presents a very interesting model that is somewhat unique in addressing non-competitive interactions between autotrophic and heterotrophic plankton via metabolites. The model is apparently very scalable and adaptable, and has been applied in the manuscript to a number of case studies - some constrained by observations, others more theoretical.

I appreciated the highly configurable nature of the model, which facilitates its application to a range of hypotheses. On the other hand, I found the model description to be very heavy going, and the use of different subscripts confusing. I felt that the schematic shown in Figure 1 could have been expanded on to reveal how the model works. In general, I did not really understand the bigger picture as I was working through the equations, and I got a bit lost in the technical details at times.

I therefore wonder if this description of the model could be made a little bit friendlier to the reader by having an initial description that explains the main concepts of the model in a less technical way. What are the state variables in the model? How do they interact in a general sense? What is configurable in terms of the model structure and its parameterisation? How would it be set up for a particular case? Etc. If the reader has access to the bigger picture earlier, it may allow them to navigate the equations a bit more easily.

Aside from this issue, I found the case studies to be illustrative, but slightly lacking in terms of rigour. In particular, in Section 4.1 experimental differences in the dynamics of phytoplankton and heterotrophs are hypothesised to be related to the exchange of organic matter and competition for nutrient. MCoM is configured accordingly, but with only one simulation it is not clear what aspects of the model are vital to reproduce the observed dynamics and which are not. There is a similar lack of any control simulations in Section 4.2. I appreciate that the main aim here is to highlight the capabilities of the model, but there is certainly room to explore these limitations in a Discussion section (which is currently not included in the manuscript).

Finally, I am not sure whether the representation of organic compounds is reasonable. In equation 14 it seems that the uptake of each compound is described by a saturating function that is independent of the concentration of all other compounds. This does not seem intuitive to me, as I would expect that uptake of related organic compounds could interfere with each other. Perhaps this is not an issue, if heterotrophs each consume only a very narrow range of organic compounds, but I feel like this issue should be examined. In this regard, it might be useful to compare to how zooplankton grazing is assumed to saturate as a function of all available prey in some models (e.g. Vallina et al. 2014).

**Specific comments**

Line 21: "Arguably the two most fundamental roles in these ecosystems are primary producers … and their counterparts, heterotrophic organisms". I would argue that primary producers and heterotrophs are actors, not roles. Roles would be primary production and heterotrophy.

Line 68: Define $t$ = time

Equation 1 and others. There is an included term for dispersion, but as far as I can see there is no mention of a spatial component in the model, aside from in equation 32, which relates to a hypothetical influx of nutrients that is not applied here. For the sake of simplicity, can the dispersion terms be removed for this version of the manuscript. It could be mentioned as a straightforward addition in a Discussion section.

Line 99: $\chi_i^N$ is mentioned with reference to equation 4, but it does not appear in that equation.

Lines 122-130: Is there any empirical support for these assumptions?

The max function in equation 9 feels a bit clunky. It is needed because some coefficients in $a_{i,j}$ are negative in order to model beneficial metabolite effects. If these beneficial effects appear elsewhere, I wonder if it would be logical to put them in a separate matrix, so that they do not enter into the mortality term (thus removing the need for the max function).

Line 230: Why are these called "pseudo-concentrations"? What are their units?

Section 3.1: The fact that large time-steps may yield inaccurate results seems general to all time-stepped models and integration schemes, so I wonder if this section could be abridged or skipped entirely for the sake of simplicity.

Figure 4: The black line is not mentioned in the legend.

Line 278: The word "polynomial" is redundant.

Figure 5: I was confused by titles in the top row, as I thought they described what was shown in the first row (which is *Synechococcus* in all cases). I would suggest "Axenic *Synechococcus*", "*Synechococcus + R. pomeroyi*" and "*Synechococcus + Tropicibacter sp.*".

Line 288: "*The decisive interaction was hypothesized to be an exchange of organic material, which provided the heterotrophs with energy and organically bound nutrient, and of remineralized nutrient, which could be assimilated by Synechococcus.*" I found this sentence quite hard to understand.

Sections 4.1 and 4.2: How were the model parameters chosen? Were the parameters formally tuned? How was model fit quantified? Can we be confident that the presented results are the best possible fit?

Line 310: : *Tropicibacter sp. … is assumed to be a strong competitor (smaller KN than Synechococcus)*" KN by itself is not sufficient to say which species is the stronger competitor for N (nutrient affinity would be). Also, "*a strong**er** competitor*"?

Line 320: Why "different possible outcomes" and not "different outcomes"?

Line 340: "are caused by the different **net** phytoplankton growth rates"?

Figure 7: This figure could be a lot clearer. In the schematic the different P, DOM and H pools should have unique subscripts (1, 2, 3) and the three time/density plots need to be labelled as it is not clear what each represents.

Line 359: "While such a setup may appear highly artificial, it is robust to parameter variations". It would be interesting to see what assumptions this behaviour is not robust to (i.e. what important mechanism breaks the pattern when removed?

Line 363: "we randomly generated communities". How? And under what constraints?

Line 410: "MCoM is intentionally kept relatively simple". The model does not come across as simple in the 11 pages and 34 equations used to describe it. I appreciate that in some ways it does have some simplicity, but this could maybe be made clearer in the way the model is described. (See also my general comments.)

**References**

Vallina, S. M., B. A. Ward, S. Dutkiewicz, and M. J. Follows. 2014. Maximal foraging with

active prey-switching: a new "kill the winner" functional response and its effect on

global species richness and biogeography. Prog. Oceanogr. **120**: 93–109.

---

## Author Comment (AC1)

**Point-by-point response to Reviewer 1**

We thank the reviewers for their thorough and positive evaluation of our manuscript and for their valuable comments, which have helped to improve this work substantially. In response, we have implemented comprehensive revisions, which we detail point-by-point below.

Please note that we have reorganized the Sections of the manuscript, relocating the content of the former Section 3, and adding a Discussion Section (Section 4). We added new control simulations for the case studies and tables of empirical ranges for parameters, which are relatively extensive. We decided to present these in a separate Supplement to the main text (supplement.pdf). We also moved parameter and notation tables from the main text to the Supplement.

**Review of "The microbial community model MCoM 1.0: A scalable framework for modelling phototroph-heterotrophic interactions in diverse microbial communities" by Lücken, Follows, Bragg and Lennartz**

**General comments**

This paper presents a very interesting model that is somewhat unique in addressing non-competitive interactions between autotrophic and heterotrophic plankton via metabolites. The model is apparently very scalable and adaptable, and has been applied in the manuscript to a number of case studies - some constrained by observations, others more theoretical.

**Comment 1:**

I appreciated the highly configurable nature of the model, which facilitates its application to a range of hypotheses. On the other hand, I found the model description to be very heavy going, and the use of different subscripts confusing. I felt that the schematic shown in Figure 1 could have been expanded on to reveal how the model works. In general, I did not really understand the bigger picture as I was working through the equations, and I got a bit lost in the technical details at times.

I therefore wonder if this description of the model could be made a little bit friendlier to the reader by having an initial description that explains the main concepts of the model in a less technical way. What are the state variables in the model? How do they interact in a general sense? What is configurable in terms of the model structure and its parameterisation? How would it be set up for a particular case? Etc. If the reader has access to the bigger picture earlier, it may allow them to navigate the equations a bit more easily.

**Response:**

Thank you for this constructive comment. We revised parts of the model description section to improve the conceptual clarity. First, we have revised Figure 1 to include a flux diagram [Panel (a)] explicitly naming the different processes, and a formal system representation using state vectors/adjacency matrices [Panel (b)]. Further, we added a new introductory

paragraph for the "Model description" section to give a conceptual overview of the modeled processes (ll.70-81).

Moreover, we structured the subsequent paragraphs into subsections, "2.1 State variables" and "2.2. Notations". Under 2.1, we list the models state variables (ll. 86-91), and under 2.2, added a note concerning the practical configuration of the program (ll. 97-100):

Further, we now explain the parameter notation in more detail (ll. 101-107).

**Comment 2:**
Aside from this issue, I found the case studies to be illustrative, but slightly lacking in terms of rigour. In particular, in Section 4.1 experimental differences in the dynamics of phytoplankton and heterotrophs are hypothesised to be related to the exchange of organic matter and competition for nutrient. MCoM is configured accordingly, but with only one simulation it is not clear what aspects of the model are vital to reproduce the observed dynamics and which are not. There is a similar lack of any control simulations in Section 4.2. I appreciate that the main aim here is to highlight the capabilities of the model, but there is certainly room to explore these limitations in a Discussion section (which is currently not included in the manuscript).

**Response:**
Thank you. It is correct that the nature of our case studies is illustrative. In this model description paper, our main intention is to demonstrate model capabilities, not to prove ecological mechanisms. However, we agree that an additional technical control may be informative. For the *Synechococcus* case study (Section 3.1 in the revised manuscript), we added simulations testing the impact of DON remineralization and DOC exudation in the Supplement (Section S3, Figure S1). We added a reference to the new Supplement section in Sec. 3.1:

"To explore the impact of DOC exudation and DON remineralization, we included comparative simulations with reduced impact of either process in the Supplement Section S3." (ll. 330-331),

and we summarize the findings and discuss the parameter fit in the first two paragraphs of the newly added Discussion section (Section 4.1, ll. 435-453).

For Section 3.2 (former Sec. 4.2 of the first submission), we included a systematic parameter sensitivity study in the Supplement Section S4. Figure S2 of the Supplement shows how *Prochlorococcus* growth curves respond to variations of key parameters (e.g., interaction strength, uptake rates). These control simulations support the hypothesis that metabolite interactions are the most plausible qualitative driver of observed differences between co-cultures with different heterotrophs, while other parameters modulate quantitative outcomes. Key insights are summarized in the Discussion:

"For Prochlorococcus co-cultures (Fig. 5), MCoM reproduced divergent phototroph dynamics through metabolite-mediated growth effects. Parameter sensitivity analysis (Supplement Section S4) demonstrated that while heterotroph traits (e.g., loss rate $\delta$ or nutrient uptake $V_N$) modulate quantitative outcomes (e.g., peak height), a variation of

interaction rates $a_M$ alone explains the qualitative transition between growth-promoting and inhibitory scenarios. This supports the hypothesis that metabolite exchange is the primary driver of experimental growth trajectory differences (Sher et al., 2011). However, we emphasize that both experiment modeling cases serve as mechanistic illustrations of MCoM's capabilities, not as validation of specific biological mechanisms underlying the experimental dynamics." (Section 4.1, ll. 454-460).

*Comment 3:*
Finally, I am not sure whether the representation of organic compounds is reasonable. In equation 14 it seems that the uptake of each compound is described by a saturating function that is independent of the concentration of all other compounds. This does not seem intuitive to me, as I would expect that uptake of related organic compounds could interfere with each other. Perhaps this is not an issue, if heterotrophs each consume only a very narrow range of organic compounds, but I feel like this issue should be examined. In this regard, it might be useful to compare to how zooplankton grazing is assumed to saturate as a function of all available prey in some models (e.g. Vallina et al. 2014).

*Response:*
Thank you. Indeed, uptake modeling is an important topic and the approach followed in MCoM certainly leaves room for refinements. Specifically concerning the comparison of bacterial and zooplankton uptake, we would like to point out that zooplankton usually has a single mode of prey-uptake leading to a joint processing of all prey types. For microbes, the uptake of different DOM compounds follows a different logic, as different compounds often require different transporter enzymes and metabolic pathways, which can decouple their processing to some degree. Independent processing pathways then saturate much like a single enzyme following Michaelis-Menten-, i.e., Monod-kinetics. This is approximate, and compounds often share metabolic machinery and can mutually influence their uptake dynamics. The conceptual compound types used in MCoM may also be understood as representing clusters of compounds, which share uptake characteristics. The functional form used in MCoM is well-established for microbial growth on mixtures of carbon sources (e.g. Mentges, 2019; Marsland, 2020a; Zakem, 2021) However, refinements are possible and we added a paragraph on this matter in the newly added Discussion section:

"Further, MCoM assumes independent saturation kinetics for organic compound uptake [Eqn. (14)], implying distinct transporter systems and metabolic pathways process each compound without interference. While this simplification aligns with established microbial substrate utilization models (e.g., Mentges et al., 2019; Marsland et al., 2020; Zakem et al., 2021), biological systems exhibit interdependencies through mechanisms of metabolic regulation or resource allocation. More complex modeling approaches incorporate such mechanisms allowing them to reproduce phenomena like sequential utilization or adaptation to substrate availability (Brandt et al., 2004; Kremling et al., 2018; Lücken and Blasius, 2025)." (ll. 484-489)

**Specific comments**
*Comment 4:*
Line 21: "Arguably the two most fundamental roles in these ecosystems are primary producers ... and their counterparts, heterotrophic organisms". I would argue that primary producers and heterotrophs are actors, not roles. Roles would be primary production and

heterotrophy.

***Response:***

We agree, that the term "roles" was not the best choice, here. We modified the sentence to:

"Arguably the two most fundamental processes in these ecosystems are photosynthetic primary production that fixes inorganic carbon and nutrients into organic biomass, and heterotrophic decomposition of organic matter back to $CO_2$ and inorganic nutrients." (ll. 21-23)

***Comment 5:***
Line 68: Define t = time

***Response:***

To clarify, that the state variables are time-dependent and that "t" is used for time (in days), we modified the sentence prior to the listing of the state variables as follows:

"In MCoM, any momentary state of the microbial model system is represented by variables describing chemical concentrations and population densities of the model's components. These state variables are time-dependent, where the time $t$ is measured in days. MCoM's state space is comprised of the following variables: [...]" (ll. 85-87)

***Comment 6:***
Equation 1 and others. There is an included term for dispersion, but as far as I can see there is no mention of a spatial component in the model, aside from in equation 32, which relates to a hypothetical influx of nutrients that is not applied here. For the sake of simplicity, can the dispersion terms be removed for this version of the manuscript. It could be mentioned as a straightforward addition in a Discussion section.

***Response:***

You are correct, the "dispersion" terms model the exchange of matter with a neighboring location. At the moment, the "dispersion" coefficient Δ was mainly used to capture chemostat setups (Sections 3.3 and 3.4; cf. parameters tables S5 and S6 in the Supplement). To avoid confusion about spatial extension we replaced the term "dispersion" by "physical transport," which captures its current semantics more exactly.

Spatially extended versions of the model are planned to be introduced in the future. We now mention this in the new Discussion section:

"As a zero-dimensional box model, MCoM lacks explicit spatial resolution. A positive transport term (coefficient Δ > 0) in the governing equations represents a bulk exchange with an external reservoir, approximating chemostat conditions and allowing continuos supply of nutrient (see case studies in Sections 3.3 and 3.4). If Δ = 0 (Sections 3.1 and 3.2), the equations describe a batch reactor, i.e., a closed system. We note that Δ could serve for future spatially explicit implementations, potentially representing diffusive connectivity in extended frameworks. Such developments would enable modeling of advective-diffusive transport and gradient-driven dynamics." (ll. 494-499)

Also, we now mention explicitly that the parameters (in particular Δ>0) for the examples in Sec. 3.3 (l. 381) and Sec. 3.4 (ll. 403-404) describe continuous nutrient supply.

***Comment 7:***
Line 99: $\chi_i^N$ is mentioned with reference to equation 4, but it does not appear in that equation.

***Response:***
Thank you for pointing this out. We removed this unnecessary mention of $\chi_i^N$.

***Comment 8:***
Lines 122-130: Is there any empirical support for these assumptions?

***Response:***
Thank you for raising this important point. The functional form governing exudate composition in response to nutrient availability and photosynthetic rate captures the observations that nutrient limitation (specifically for nitrogen and phosphorus) promotes carbon-rich DOM exudation, while nutrient-replete conditions favor nitrogen-rich DOM release. Our mathematical formulation follows a decoupling of uptake and growth rates. We acknowledge that this implementation represents a simplified formulation rather than a direct derivation from empirical data.

Our formulation was designed to allow flexibility to accommodate observed variability: users can parametrize both the stoichiometric ratio (C:N) of exuded DOM and the fraction of fixed carbon allocated to exudation, or disable the mechanism entirely. We mention the general pattern of nutrient-dependent DOM composition in the Introduction:

"The composition of these exudates can vary significantly depending on environmental conditions, phytoplankton species, and growth phase (Buchan et al., 2014). In particular, the elemental ratios of the exudates are influenced by the growth-limiting factor, resulting in varying carbon-to-nutrient ratios, potentially imposing subsequent limitation on heterotrophic bacterial communities (Saad et al., 2016)." (ll. 34-37)

Further, we have added a paragraph in the Discussion, which hopefully clarifies this:

"An important assumption in MCoM concerns the parameterization of DOM exudation mechanics. The fundamental nutrient-dependent stoichiometric pattern, where phototrophs exude carbon-rich compounds under nutrient limitation and nitrogen-rich compounds under nutrient sufficiency, is empirically well-established (Myklestad, 2000; Thornton, 2014). However, the precise functional form governing the extent and composition of exudates in MCoM represents a tractable implementation rather than a formulation derived from species-specific empirical data. Our approach provides flexibility through user-defined parameters ($r^{C:N}_{i \to ex}$ and $q^{max}_i$) to accommodate some variability, while explicitly enforcing biochemical constraints (e.g., nitrogen-rich exudates require carbon backbones). Since the mechanistic links between nutrient availability, photosynthetic rate, and exudation dynamics remain an active research area (Saad et al., 2016; Wu et al., 2021), caution is warranted when making quantitative interpretations." (ll. 473-481)

*Comment 9:*
The max function in equation 9 feels a bit clunky. It is needed because some coefficients in a_{i,j} are negative in order to model beneficial metabolite effects. If these beneficial effects appear elsewhere, I wonder if it would be logical to put them in a separate matrix, so that they do not enter into the mortality term (thus removing the need for the max function).

*Response:*
Thank you for this constructive suggestion. Multiple approaches exist for implementing metabolite impacts on microbial population dynamics. The most immediate alternative to eliminate the maximum function in Eq. (9) would involve multiplicative modulation of growth or mortality through non-negative factors (e.g., an inverse sigmoid function of metabolite concentration). However, this alternative does not substantially improve simplicity, as it would introduce additional parameters to characterize the modulator function, thereby increasing model complexity.

MCoM's current formulation is predicated on the principle that metabolites with positive impacts reduce mortality rates, while those with negative impacts increase mortality. A conceptually similar approach could alternatively model these effects on uptake or photosynthesis rates rather than mortality. However, this would similarly necessitate maximum functions to preserve the directionality of these processes.

We acknowledge that the maximum function may lack mathematical elegance but maintain that it represents the most pragmatic implementation given current uncertainties regarding the precise functional form of metabolite impacts.

*Comment 10:*
Line 230: Why are these called "pseudo-concentrations"? What are their units?

**Response:**
Thank you for this question. The concentrations have units of 1/L (as defined in line 92), which differs from other model components – e.g., DOC components are in carbon-molar units (µmol C/L). As metabolites are assumed to occur and be effective in spurious concentrations, their numerical unit is kept abstract for simplicity, relieving the user from defining additional stoichiometric parameters, where their impact is negligible. Biological effects are mediated through equally abstract interaction rate coefficients ($a_{j \to i}$ in units 1/day) that absorb scaling factors.

The "pseudo" designation originally acknowledged this intentional omission of units where concentrations represent relative bioactive potentials rather than absolute chemical inventories. We agree the term may cause confusion and have removed the 'pseudo' prefix in the revised manuscript. The $M_j$ are now simply referred to as "metabolite concentrations".

*Comment 11:*
Section 3.1: The fact that large time-steps may yield inaccurate results seems general to all time-stepped models and integration schemes, so I wonder if this section could be abridged or skipped entirely for the sake of simplicity.

*Response:*

Thank you for this suggestion. We agree that the caution regarding time-step sensitivity represents a general consideration for numerical integration rather than a model-specific feature. Consequently, we have removed this remark to streamline the text.

As part of broader manuscript restructuring to accommodate the new Discussion section, we have reorganized the original Section 3 content:

- The technical details of the integration scheme (formerly Section 3.1) now appear in Supplement Section S8.

- The performance analysis (former Section 3.2) has been transferred into the Discussion section (now Section 4.3; ll. 505-515)

*Comment 12:*

Figure 4: The black line is not mentioned in the legend.
Line 278: The word "polynomial" is redundant.

*Response:*

Thank you for noting this. We moved the explanation of the black curve from the main text into the caption of the figure, omitting the term "polynomial". Note that the figure has been moved to the Discussion and is now Figure 8.

*Comment 13:*

Figure 5: I was confused by titles in the top row, as I thought they described what was shown in the first row (which is Synechococcus in all cases). I would suggest "Axenic Synechococcus", "Synechococcus + R. pomeroyi" and "Synechococcus + Tropicibacter sp.".

*Response:*

Thank you, the suggested titles are clearer, indeed. We updated Figure 4 (former Figure 5) accordingly.

*Comment 14:*

Line 288: "The decisive interaction was hypothesized to be an exchange of organic material, which provided the heterotrophs with energy and organically bound nutrient, and of remineralized nutrient, which could be assimilated by Synechococcus." I found this sentence quite hard to understand.

*Response:*

We broke up this sentence into shorter ones to make the passage more readable:

"The key interaction was hypothesized to be a mutual exchange of essential resources. *Synechococcus* provided organic material to the heterotrophs, from which these extracted energy and nutrients. In return, the heterotrophs remineralized nutrients, making them available to *Synechococcus* again." (ll. 299-302)

*Comment 15:*

Sections 4.1 and 4.2: How were the model parameters chosen? Were the parameters formally tuned? How was model fit quantified? Can we be confident that the presented results are the best possible fit?

*Response:*

We thank the reviewer for raising this methodological consideration. We emphasize that the primary aim of the case studies in Sections 4.1 and 4.2 is to illustrate the capabilities of MCoM in reproducing microbial interactions in experimental contexts. Parameters for both case studies were initially chosen manually within biologically plausible ranges derived from the literature. To document this, and to provide a general guidance for choosing parameters, we now provide tables in the Supplement (Tables S7-S17) for key parameters, where empirical values are available.

For modeling the *Synechococcus* experiments (Section 4.1), manual tuning aimed at reproducing approximate timing and amplitude of population growth phases. For the revised version, we have implemented an optimization procedure, which uses a Nelder-Mead algorithm to automatically maximize the R² values of the simulated growth curves over a subset of parameters. We describe this in Section 3.1:

"We modelled both co-cultures and the axenic growth using MCoM with fixed growth parameters for the phototroph and specific characteristics for the heterotrophs. We first selected all parameters manually within plausible ranges (cf. Supplement Section S7), aiming at a qualitative reproduction of the timing and amplitude of the observed changes in cell densities. Using this parametrization as a starting point, a subset of parameters was used for an automatic maximization of the sum of the $R^2$-values for the different experiments (using the Nelder-Mead routine of the scipy.optimization package, Virtanen et al., 2020). Optimized parameters are: $V_{N \to i}$, $f^{max}_{C \to i}$, and $\delta_i$ for *Synechococcus*, and $V_{DON \to i}$, $V_{DOC \to i}$, $V_{DON \to i}$, $V_{DON \to i}$, and $\delta_i$ for the heterotrophic bacteria. The exact values are listed in Supplement Table S3. Importantly, we did not assume metabolite interactions, but the observations could be reproduced qualitatively by the exchange of nutrients in inorganic and organic form." (ll. 310-317)

While revisiting the *Prochlorococcus* experiments (Section 4.2), we discovered a mistake in initial DOM concentration for our simulations (we used mM instead of µM). For the updated fit, after a manual approximation, we similarly re-calibrated the parameters through a formal Nelder-Mead optimization of selected parameters (including the scale for comparing measured fluorescence units with simulated cell numbers). This is described in Section 3.2:

"All parameters were initially selected manually within plausible ranges. Subsequently, key parameters ($V_{N \to H}$ , $V_{N \to P}$ , $\delta_P$ , $V_{DOC \to H}$ , $Y_{DOC \to H}$, $a_M$) were subjected to an automatic minimization (using the Nelder-Mead routine of the scipy.optimization package, Virtanen et al., 2020) of the root mean square error of simulated and observed growth curves." (ll. 344-347)

***Comment 16:***
Line 310: : Tropicibacter sp. ... is assumed to be a strong competitor (smaller KN
than Synechococcus)" KN by itself is not sufficient to say which species is the stronger
competitor for N (nutrient affinity would be). Also, "a stronger competitor"?

***Response:***
> Thank you. This is correct, of course. We made this statement more accurate, following
> your suggestion:
> "[…] is assumed to have a higher nutrient affinity ($V_N/K_N$) than *Synechococcus*, i.e., a
> competitive advantage under low-nutrient conditions." (ll. 328-329)

***Comment 17:***
Line 320: Why "different possible outcomes" and not "different outcomes"?

***Response:***
> You are correct, "possible" is redundant here. We removed it in the revised manuscript.

***Comment 18:***
Line 340: "are caused by the different net phytoplankton growth rates"?

***Response:***
> Yes, adding "net" makes the statement more precise. Thank you for the suggestion. We
> added it in the revised version.

***Comment 19:***
Figure 7: This figure could be a lot clearer. In the schematic the different P, DOM and H pools
should have unique subscripts (1, 2, 3) and the three time/density plots need to be labelled as it
is not clear what each represents.

***Response:***
> Thank you for pointing this out. We have now added subscripts to the different components
> in Panel (a) of Figure 6 (former Figure 7) and moved the corresponding parts of the legend
> into the different subpanels of (b).

***Comment 20:***
Line 359: "While such a setup may appear highly artificial, it is robust to parameter variations". It
would be interesting to see what assumptions this behaviour is not robust to (i.e. what
important mechanism breaks the pattern when removed?

***Response:***
> Thank you, this is an interesting aspect, indeed. In principle, our system extends the
> classical cyclic dominance framework (May and Leonard, 1975) to phytoplankton-bacteria
> consortia. The crucial property of these systems is a "rock-paper-scissors" relationship
> between the competitors: each consortium must have both a superior competitor (positively
> affected by its metabolites) and an inferior competitor (providing beneficial metabolites to
> it). While for the classical Lotka-Volterra model limit cycles appear only at very specific
> parameter combinations, structurally stable oscillations emerge in resource competition
> models (Huisman and Weissing, 1999). Without providing mathematical rigor, we suggest
> to understand the dynamics observed in the cyclic MCoM motif in that line.

We added a section to the Supplement (Section S5), exploring the impact of two parameters (maximal nutrient uptake rate $V_{N \to 1}$ of one species and interaction strength $a$) on the dynamics. We observe that imbalances in the nutrient uptake rates disrupt the non-hierarchical relationship. Whether weakened or strengthened, a significantly distinct uptake in one species leads either to its dominance (for elevated uptake), or the dominance of the strongest remaining competitor (for diminished uptake), see Fig. S3. Second, when metabolite interaction strength is weakened, baseline uptake differences dominate, establishing competitive hierarchies rather than cyclic advantage (Fig. S4). We added a reference to the Supplement section in Section 3.3 of the main text:

"While such a setup may appear highly artificial, it is robust to parameter variations (see Supplement Section S5) and illustrates the potential of metabolite feedbacks to incite non-stationarity of population densities even if no environmental forcing is present." (ll. 383-385).

**Comment 21:**
Line 363: "we randomly generated communities". How? And under what constraints?

**Response:**
Thank you for this question. We revised the paragraph describing the random generation to clarify this (modified text in blue):

"For each simulated community, two consortia, A and B, were generated, each consisting of ten phototrophs (distinct in their specific photosynthesis characteristics), ten heterotrophs, ten DOC and ten DON compounds, and ten metabolite types. For the generation, we used MCoM's random community generation facilities to generate two consortia (see the README in the source code repository for details). As input for the generation, we prescribed the in- and out-degrees of the microbial nodes, i.e., the number of released and consumed compounds, the number of produced and effective metabolites, and the ranges, from which photosynthesis parameters for the single phototrophs were drawn (cf. Supplement Table S6). MCoM's algorithm then randomly connects the different components adhering to these degrees. Subsequently, we coupled the two consortia by defining a number of shared metabolites, which effect both consortia. This number is called "overlap" in the following. Figure 7 (a) shows the community structure schematically." (ll. 390-398)

In particular, we now added a reference to the README file accompanying the source code, where all details of randomized community generation are described.

**Comment 22:**
Line 410: "MCoM is intentionally kept relatively simple". The model does not come across as simple in the 11 pages and 34 equations used to describe it. I appreciate that in some ways it does have some simplicity, but this could maybe be made clearer in the way the model is described. (See also my general comments.)

**Response:**
You are right, there might be an emphasis on "relatively". It is simple relative to its complexity in the sense that each component and the interaction between the different components is simple. For instance, microbial populations do not include adaptive

stoichiometry, reserve dynamics, metabolic adaptation, or environmental adaptation, neither does MCoM differentiate between mortality and maintenance. Further, treating DOC and DON compounds separately effectively neglects compound stoichiometry and, as noted above, more complex functional forms for uptake responses are possible.

We modified the sentence to clarify that the simplicity is attributed to individual model components:

" Although MCoM can be used to model complex communities, which exhibit rich dynamics, the individual interactions and growth dynamics are intentionally kept relatively simple […]" (ll. 527-528)

---

## Author Comment (AC2)

**Point-by-point response to Reviewer 2**

We thank the reviewers for their thorough and positive evaluation of our manuscript and for their valuable comments, which have helped to improve this work substantially. In response, we have implemented comprehensive revisions, which we detail point-by-point below.

Please note that we have reorganized the Sections of the manuscript, relocating the content of the former Section 3, and adding a Discussion Section (Section 4). We added new control simulations for the case studies and tables of empirical ranges for parameters, which are relatively extensive. We decided to present these in a separate Supplement to the main text (supplement.pdf). We also moved parameter and notation tables from the main text to the Supplement.

**Summary**

Lücken and colleagues present a 0D box model of phototroph-heterotroph-DOM-nutrient interactions. The paper is simple and easy to read and understand. The model equations and motivations are well stated. The outcomes are clearly drawn via use cases against some experimental data.

I have minor comments given that the paper is well written, well presented, and doesn't overreach. It is a simple model, and the authors are able to fit/explain some experimental data with phototroph-heterotroph interactions with only simple tweaks to their model parameters. While these explanations are not deeply convincing, it does convince me that the model is useful for exploratory studies.

My only comments are:

- Line 79: Can you please define X and Y sub- super-scripts and what these are referring to exactly. Is X flux in and Y flux out?

*Response:*

Thank you for that suggestion. We have modified the Model description section, in particular the initial part, to provide a clearer overview of the model and the notations. Regarding the notation for fluxes, we have changed the notation to $f_{X \to Y}$ throughout the manuscript. Here, $X$ denotes the source and $Y$ the sink of the flow. We state this unambiguously in the revised manuscript:

"For the formal presentation, we use the notation $f_{X \to Y}$ to denote elemental fluxes, where $X$ is the source and $Y$ the sink of the corresponding flow." (ll. 103-104)

- Line 308: While this is one way to explain the data, it's not super convincing. However, you are also not saying that this result is the ultimate explainer… So, I think that's fine. Perhaps just make a note here that there are other possible processes/mechanisms that could explain this.

*Response:*

We agree that modeling the accumulation of all inaccessible nitrogen solely as refractory dissolved organic nitrogen (DON) may not fully capture the reality. In our configuration, the refractory DON pool only serves as a conceptual representation for diverse irreversible nutrient losses. We have added a sentence in the revised manuscript to clarify this:

"This pool represents a sink for non-recyclable matter, encompassing not only refractory DON, but all other inaccessible nutrient sinks, e.g., particulate forms." (ll. 325-326)

- Line 334: by reducing loss rates right?

*Response:*

Yes, that's correct. To make this clear, we modified the sentence to:

"For the inhibitory case, metabolite effects increase the phytoplankton loss rate, resulting in a reduced net growth rate." (ll. 355-356)

- Figure 8, panel f: can you give a more clear explanation of this figure for the reader in the text? It took me some time to figure out exactly what was going on here. Your explanation is already okay, but perhaps too concise to allow for smooth interpretation.

*Response:*

Thank you for this comment. We included additional explanations for Figure 7(f) [former Figure 8(f)] in the main text:

"The relative averaged abundance over the last ten years of the simulation is reported for each simulation run in Panel (f). In this panel, for each overlap value, two horizontal bar groups are displayed: the top group corresponds to initializations with consortium A dominant, and the bottom group to consortium B dominant. Each group contains 20 stacked bars (one per community), where each bar represents 100% relative abundance. The bars are divided into colored segments: shades of green for species from consortium A and shades of red for consortium B, with segment lengths proportional to species' average abundances. Starting from different initial states, priority effects appear as systematic differences in the dominant consortium (i.e., predominantly green vs. red segments) between the top and bottom groups for the same overlap. Convergence is indicated when both groups exhibit similar color distributions." (ll. 418-425)

Further, we extended the description of Panel (f) in the figure's caption, now reading:

" (f) Asymptotic relative abundances by consortium (green: A, red: B) for different overlap values and initial states. For each overlap, top bar group: initial A dominance; bottom group: initial B dominance. Each horizontal bar represents one community's species distribution (n = 20 communities per group)."